# *Scenethesis*: A Language and Vision Agentic Framework for 3D Scene Generation

**Lu Ling**[1,2]    **Chen-Hsuan Lin**[1]    **Tsung-Yi Lin**[1]    **Yifan Ding**[1]    **Yu Zeng**[1]

**Yichen Sheng**[1]    **Yunhao Ge**[1]    **Ming-Yu Liu**[1]    **Aniket Bera**[2*]    **Zhaoshuo Li**[1*]

[1]NVIDIA Research
[2]Purdue University
https://research.nvidia.com/labs/dir/scenethesis/

## ABSTRACT

Generating interactive 3D scenes from text requires not only synthesizing assets but also arranging them with spatial intelligence—support, affordances, and plausibility. However, training data for interactive scenes is dominated by a few indoor datasets, so learning-based methods overfit to in-distribution layouts and struggle to compose diverse arrangements (e.g., outdoor settings and small-on-large relations). Meanwhile, LLM-based layout planners can propose diverse arrangements, but the lack of visual grounding often yields implausible placements that violate commonsense physics. We propose *Scenethesis*, a training-free, agentic framework that couples LLM-based scene planning with vision-guided layout refinement. Given a text prompt, *Scenethesis* first drafts a coarse layout with an LLM; a vision module refines the layout and extracts scene structure to capture inter-object relations. A novel optimization stage enforces pose alignment and physical plausibility, and a final judge verifies spatial coherence and triggers targeted repair when needed. Across indoor and outdoor prompts, *Scenethesis* produces realistic, relation-rich, and physically plausible 3D interactive scenes, reducing collisions and stability failures compared with SOTA methods, making it practical for virtual content creation, simulation, and embodied AI.

## 1 INTRODUCTION

Synthesizing interactive 3D scenes from text is crucial for gaming (Hu et al., 2024), virtual content creation (Öcal et al., 2024), and embodied AI (Yang et al., 2024c;b; Deitke et al., 2023a; Kolve et al., 2017; Krantz et al., 2020; Nasiriany et al., 2024). Beyond asset synthesis, realistic scenes must go beyond photorealism to support spatial intelligence—reasoning about support, occlusion, and affordances in editable, physically coherent environments—so objects serve functional roles and afford interaction. Traditional interactive scene generation methods, including manual design (Kolve et al., 2017; Gan et al., 2020; Li et al., 2023a) and procedural approaches (Deitke et al., 2022), are either labor intensive or overly simplified inter-object relations.

Recent learning-based approaches, including layout generation (Paschalidou et al., 2021; Yang et al., 2024b; Tang et al., 2024) and multi-instance 3D scene generation (Huang et al., 2025a), learn object arrangement directly trained on CAD dataset such as 3D-FRONT (Fu et al., 2021). However, both families inherit the biases of the available indoor datasets: they are predominantly indoor, skewed toward large furniture, and even contain collisions (Yang et al., 2024b), leaving *small/supporting objects* and *long-tail* relations (on-top-of / inside / behind) underrepresented. Consequently, evaluations emphasize in-distribution synthetic splits, and generalization to *out-of-distribution*(OOD) layouts—outdoor scenes, rare support relations, and small objects—remains weak.

Meanwhile, LLM-based layout planners expand semantic diversity by leveraging textual commonsense (Yang et al., 2024c; Feng et al., 2024; Kumaran et al., 2023), but ungrounded planning often yields placements that violate object function roles and basic physical constraints (e.g., misoriented

---

*co-last author.

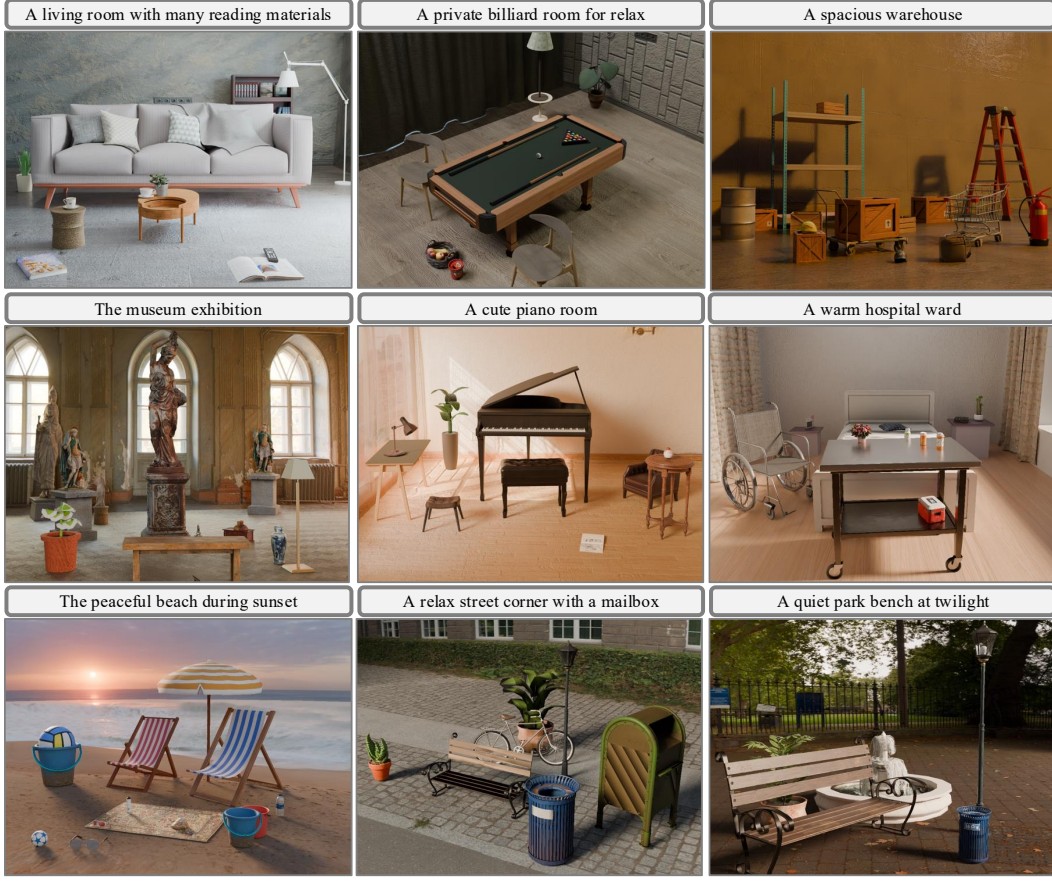

Figure 1: Given a text prompt, *Scenethesis* leverages both language and visual priors to generate realistic and physically plausible indoor and outdoor environments.

seating and blocked openings). Such misplacements break functional affordances and spatial coherence, reducing the scene to a collection of objects rather than a coherent layout, and thereby undermining downstream usability.

Motivated by vision foundation models that encode compact real-world spatial priors, we introduce *Scenethesis*, a training-free, agentic framework that integrates LLM planning with vision-guided spatial refinement. Given a prompt, the LLM drafts a coarse layout; the vision module refines layout and extracts scene structure to capture inter-object relations. A lightweight optimization then aligns object poses and enforces physical plausibility via semantic correspondence and SDF-based contact/support constraints, preventing interpenetration and instability. A final judge verifies spatial coherence and triggers re-planning when needed. This design preserves the open-ended diversity of language planning while injecting the visual grounding and physics awareness absent from LLM-based pipelines without training a new scene-level generator. Figure 1 shows examples of indoor / outdoor scenes generated by *Scenethesis*. Our contribution is summarized as follows.

- We introduce *Scenethesis*, a training-free, agentic framework that integrates LLM planning, vision foundation models, physics-aware optimization, and a judge module to synthesize *open-ended, physically plausible* interactive scenes.
- We propose an efficient physically plausible optimization guided by semantic correspondences and SDF-constraints, enforcing collision-free and stable placements without post-hoc simulation.
- On benchmarks spanning indoor/outdoor prompts and long-tail spatial relations (on-top-of, inside, behind), *Scenethesis* consistently improves layout realism and physical plausibility relative to state-of-the-art methods, producing diverse scene layouts for downstream use.

## 2 RELATED WORK

**Indoor scene synthesis.** Early work cast scene synthesis as layout prediction with relation graphs or hierarchical structures (Luo et al., 2020; Chang et al., 2014; Zhou et al., 2019; Li et al., 2019; Wang et al., 2019). Learning-based layout generation methods model layout via auto-regression and diffusion (Wang et al., 2021; Paschalidou et al., 2021; Tang et al., 2024; Lin & Mu, 2024; Yang et al., 2024b). Multi-instance scene generation methods leverage strong image-to-3D priors to output assets and poses in one pass (Huang et al., 2025a; Meng et al., 2025; Lin et al., 2025a) to post-optimization. Despite these advances, both families inherit biases from predominantly indoor datasets (e.g., 3D-FRONT/3D-FUTURE (Fu et al., 2021) ): emphasis on large furniture and under-represent *small/supporting* objects and *long-tail* relations (on-top-of / inside / behind). Consequently, generalization to *OOD* layouts—outdoor scenes and rare support relations underexplored. The recent learning-based approach I-Scene (Ling et al., 2025) exhibits strong generalization, extending scene generation from structured indoor layouts to complex outdoor environments with rich spatial relationships, while not explicitly encoding physical constraints in the training objective. In contrast, *Scenethesis* tackles *text-to-scene* with a training-free, physics-aware optimization loop, aiming for open-ended layout diversity and physically plausible arrangements beyond indoor distributions.

**LLM/VLM Guided 3D Scene Generation.** With the advancement of LLMs/VLMs, recent methods (Öcal et al., 2024; Yang et al., 2024c; Çelen et al., 2024; Kumaran et al., 2023; Zhou et al., 2024c; Wang et al., 2023; Aguina-Kang et al., 2024; Lin et al., 2024a; Sun et al., 2024; Zhou et al., 2024b) use language models for: (1) spatial relation planning, (2) 3D asset retrieval from semantic descriptions or vision-language embeddings, and (3) rule-based rough collision detection. However, LLM-based planning is ungrounded: it operates in symbolic space rather than metric geometry, and thus often misorients furniture, violates support/clearance constraints, and produces floating or intersecting placements—especially for small or occluded objects. Template relations in text are coarse and miss room-specific structure (Lin & Mu, 2024; Khanna et al., 2024) In contrast, *Scenethesis* uses LLM priors only to draft coarse layout instructions, then grounds it with a vision foundation model to preserve compact spatial information, effectively capturing real-world spatial complexity.

**Visual Foundation Model-Guided Scene Generation.** Visual-foundation-model pipelines often synthesize 3D scenes by optimizing a single global representation (e.g., NeRF/3DGS) from 2D priors (Höllein et al., 2023; Fridman et al., 2024; Yu et al., 2024b;a; Zhang et al., 2024a). While photorealistic, these monolithic approaches lack instance structure, hindering per-object editing and interaction. A second line composes scenes via perception → per-object synthesis → geometric fitting (Wang et al., 2024; Zhou et al., 2024a; Dai et al., 2024; Huang et al., 2025b; Rahamim et al., 2024); CAST further adds *post-hoc* physics-aware pose correction to reduce penetration and floating artifacts (Yao et al., 2025). Instead of a post-hoc physical correction stage, *Scenethesis* is a text-to-scene and training-free, enforcing contact/support constraints inside the pose-alignment loop with a judge-and-repair agent—so physics shapes the layout as it is formed, not patched after the fact.

**Physics-Aware Scene Generation.** Physical principles are often underexplored in interactive scene generation. Most methods approximate geometry with 3D bounding boxes (Yang et al., 2024b;c), a coarse proxy that breaks small-on-large relations and causes floating and interpenetration (Çelen et al., 2024). CAST adds physical correction but applies it post-hoc (Yao et al., 2025). In contrast, *Scenethesis* performs mesh-level collision tests by sampling points on object surfaces and injects SDF-based contact/support constraints *in-loop*, substantially reducing collisions and instability.

## 3 METHOD

*Scenethesis* generates **spatially realistic, physically plausible** interactive 3D environments from user prompts. An overview of the pipeline is shown in Figure 2, consisting of four key stages: (1) an **LLM module** drafts a coarse scene plan, (2) a **vision module** refines the layout with visual guidance and structural extraction, (3) a **physical-aware optimization module** distills priors and adjusts object placement for spatial coherence and physical plausibility, and (4) a **scene judge module** verifies spatial consistency. The following sections detail each module's role.

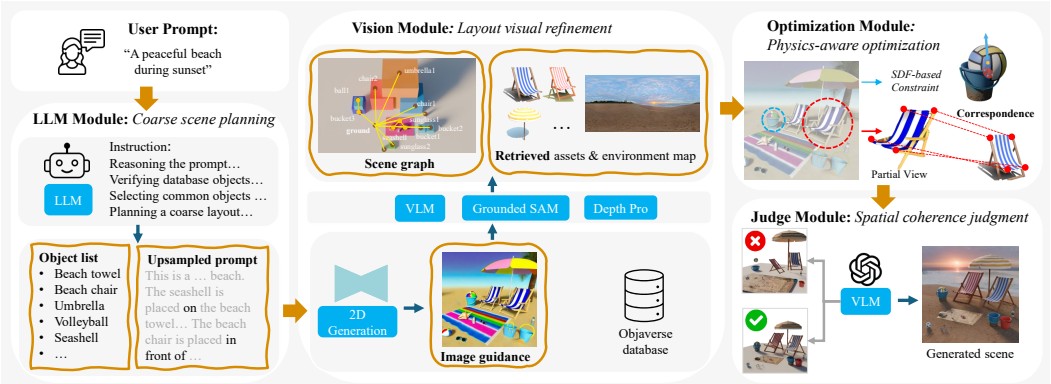

Figure 2: *Scenethesis* is an agentic pipeline: an LLM drafts a coarse scene planning, a vision module grounds and refines it, a physics-aware optimizer iteratively aligns poses and enforces contact/support constraints, and a judge verifies spatial coherence.

## 3.1 COARSE SCENE PLANNING

*Scenethesis* supports either a **simple prompt** (e.g., "a peaceful beach during sunset") for flexible scene generation or a **detailed prompt** for controllable scene generation (e.g., a scene plan describing the detailed spatial relations as shown in the *appendix*). For a simple prompt, the LLM generates a coarse scene plan by reasoning over user input. It first interprets the prompt, reviews all object categories in the 3D database, selects commonly associated objects, and then generates an up-sampled prompt describing coarse spatial relations, as illustrated in Figure 2. When given detailed prompts, the LLM checks for the presence of all specified objects in the database, infers relevant object categories, and skips the prompt up-sampling process.

Following prior work (Yang et al., 2024c), the LLM selects an *anchor* object (the top of the scene hierarchy after the *ground*) and builds a coarse spatial hierarchy by placing other objects relative to this anchor. These relations are folded into an upsampled prompt. Example: in a cozy living room, the sofa is the anchor at the *center*; a bookshelf sits in the *background* aligned to the wall; a coffee table and chairs are placed *in front of* or *beside* the sofa.

## 3.2 LAYOUT VISUAL REFINEMENT

*Scenethesis* exploits the spatial priors implicit in image generators: large-scale training imbues them with common co-occurrences and object arrangements. We use these priors to refine the LLM draft via three steps: (1) *Image Guidance* – Generates images to refine spatial relations, ensuring realism and object functionality. (2) *Scene Graph Generation* – Segments objects, estimates depth and 3D bounding boxes (3DBB), and constructs a graph encoding inter-object relationships. (3) *Asset Retrieval* – Selects 3D assets and environment maps for final scene composition.

**Image Generation.** The **vision module** refines the upsampled prompt into a visually grounded image. This generated image serves as the basis for segmentation, depth estimation, and asset retrieval.

**Scene Graph Generation.** Using VFMs/VLMs (e.g., GPT-5 (OpenAI, 2025), Grounded-SAM (Ren et al., 2024), DepthPro (Bochkovskii et al., 2024)), we construct a scene graph with 3DBBs and identify structural components, including the *anchor*, *parent*, and *child* objects (see Figure 2). We initialize 5-DoF poses by segmenting each object, lifting with monocular depth, and back-projecting to a sparse 3D point cloud. Because occlusions, limited viewpoints, and segmentation noise can bias 3DBBs, these poses are only coarse and are adjusted in the optimization stage (Section 3.3).

The scene graph forms the basis for iterative 5DoF pose adjustments during optimization in the next stage. Since *Scenethesis* targets ground-level layout, wall-mounted/background elements are provided by the retrieved environment map. Details of the scene-graph format appear in the *appendix*.

**Asset Retrieval.** Generative/reconstruction pipelines (e.g., 3DGS-based (Liang et al., 2024; Wu et al., 2024)) can be photorealistic but often introduce geometric artifacts and lack edit-ready outputs

(manifold meshes, UVs, decomposable PBR), limiting downstream use. We therefore adopt retrieval: *Scenethesis* selects category/attribute-matched assets from a curated, high-quality Objaverse subset (Deitke et al., 2023b) (following (Yang et al., 2024c)) and a custom environment-map set. These assets and an environment map are retrieved to assemble the scene, providing both geometric fidelity and editability. Retrieval and filtering details are in the *Appendix*.

### 3.3 PHYSICS-AWARE OPTIMIZATION

Directly placing 3D assets on 3DBBs estimated from image guidance poses significant challenges: (1) *Occlusions* yield incomplete point clouds, corrupting orientation, scale, and position and thus degrading 3D-BB accuracy. (2) *Asset–image mismatch* (shape/texture differences between retrieved assets and the guidance image) hampers precise pose estimation. To overcome these issues, *Scenethesis* introduces a **physics-aware optimization** powered by robust semantic feature matching and signed-distance fields (SDFs) (Zhang et al., 2024b; Edstedt et al., 2024; Chen et al., 2024a). This optimization process iteratively refines object poses to ensure pose alignment and physical plausibility.

#### 3.3.1 POSE ALIGNMENT

To mitigate pose errors from occlusion, segmentation noise, and asset mismatch, we use dense, semantics-aware correspondences from RoMa (Edstedt et al., 2024), which are robust to partial views. Let there be $N$ objects in the rendered image $I$, where object $\mathbf{o}_i$ has a 5-DoF configuration (scale, upright rotation, translation) and its counterpart in the guidance image $\tilde{I}$ is $\tilde{\mathbf{o}}_i$. We extract $m$ high-confidence correspondences between the two:

$$\{\, p(x,y),\, \tilde{p}(x,y) \,\}_i^m \;=\; \mathrm{RoMa}(\mathbf{o}_i,\, \tilde{\mathbf{o}}_i)\,, \tag{1}$$

where $p(x,y)$ and $\tilde{p}(x,y)$ are corresponding points on $\mathbf{o}_i$ and $\tilde{\mathbf{o}}_i$, respectively. We then optimize the 5-DoF parameters by minimizing a weighted sum of 2D reprojection and 3D consistency over these correspondences, and backpropagate gradients to refine scale, translation, and upright rotation (see Figure 2). Full details appear in the *Appendix*.

#### 3.3.2 PHYSICAL PLAUSIBILITY

Real-world 3D scenes obey physical laws, ensuring objects remain stable on contact surfaces and collision-free. However, pose alignment with image guidance alone does not guarantee physical plausibility—objects may intersect, float, or sink due to imperfect scene understanding. See Figure 7 (b) as an example.

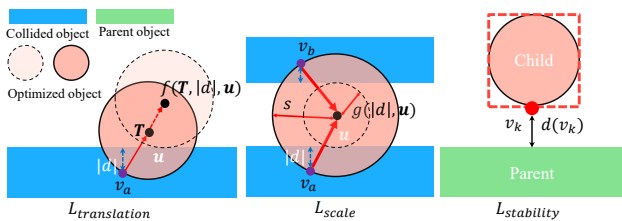

Figure 3: *Collision avoidance* and *stability maintenance*.

Prior work often approximates objects with 3D bounding boxes (3DBBs) (Yang et al., 2024c;b). This coarse proxy ignores shape, making *containment/under* relations impossible and distorting inter-object constraints—e.g., items cannot be placed inside shelf compartments because 3DBBs falsely collide (see Figure 6). We instead operate on surface geometry: for each object we sample points on the mesh and use them for precise collision tests and SDF-based contact/support reasoning, enabling accurate placement (*inside/under/on*).

The **physics-aware optimization** iteratively constructs a SDF-based physical structure, following the scene graph hierarchy: processing the anchor object first to establish a stable foundation, followed by parent and child objects. The physics-aware optimization incorporates **collision** and **stability constraints**. Since retrieved 3D assets are upright, their rotation is constrained to azimuthal adjustments.

Formally, given a scene graph with $N$ objects, each object has a 5-DoF configuration defined by scale $s$, upright rotation $\mathbf{R}$, and translation $\mathbf{T} = (t_x, t_y, t_z)$. For computational efficiency, we uniformly sample $n$ points from its triangle surface mesh as its geometric representation and compute its centroid for collision avoidance.

Table 1: Quantitative evaluation on text–image alignment and spatial quality (↑ higher is better). Spatial quality preference measures GPT-5 and human preference for **Ours** over baselines.

| Method | Text–Image Alignment | | | Spatial Quality Preference (GPT-5 / Human Evaluation) | | | |
|---|---|---|---|---|---|---|---|
| | CLIP↑ | BLIP↑ | VQA↑ | Object Diversity↑ | Layout Coherence↑ | Spatial Realism↑ | Overall Performance↑ |
| MIDI | – | – | – | 50% / 55% | 60% / 65% | 75% / 82% | 70% / 78% |
| Digital Cousins | – | – | – | 55% / 55% | 80% / 95% | 80% / 94% | 80% / 92% |
| DiffuScene | 23.11 | 48.28 | 0.7832 | 75% / 80% | 80% / 90% | 90% / 76% | 80% / 80% |
| SceneTeller | 25.27 | 51.99 | 0.7999 | 80% / 85% | 80% / 71% | 85% / 80% | 80% / 74% |
| Holodeck | 28.32 | 46.25 | 0.6815 | 85% / 80% | 83% / 78% | 81% / 86% | 85% / 85% |
| LayoutGPT | 23.01 | 46.35 | 0.8052 | 85% / 85% | 80% / 71% | 90% / 82% | 90% / 71% |
| LayoutVLM | 24.57 | 41.96 | 0.6365 | 80% / 85% | 80% / 90% | 80% / 85% | 85% / 85% |
| IDesign | 28.19 | 44.76 | 0.7095 | 60% / 65% | 70% / 74% | 70% / 72% | 65% / 77% |
| **Ours** | **30.71** | **77.17** | **0.8269** | – / – | – / – | – / – | – / – |

**Collision Constraints.** We query the scene SDFs using object surface points to detect collision states and define position collision loss $\mathcal{L}_{\text{translation}}$ and scale collision loss $\mathcal{L}_{\text{scale}}$. As shown in Figure 3, the deviation caused by collisions impacts translation $T$ as:

$$\mathcal{L}_{\text{translation}} = \sum_{\mathbf{v}_i \in \mathbf{V}^-} ||f(\mathbf{T}, |d_i|, \mathbf{u}_i) - \mathbf{T}||_2^2, \tag{2}$$

where $f(\mathbf{T}, |d_i|, \mathbf{u}_i) = \mathbf{T} + \mathbf{u}_i \cdot |d_i|$ computes a collision-free position by adjusting the translation along direction $\mathbf{u}_i$ with step size $|d_i|$. Here, $d_i$ is the negative SDF value at a collided point $\mathbf{v}_i$, which belong to the points set with negative SDF $\mathbf{V}^-$ sampled uniformly from the surface. The direction $\mathbf{u}$ is defined from the collision point toward the model's centroid, guiding objects away from collisions.

Collisions also affect object scale $s$ due to opposing forces:

$$\mathcal{L}_{\text{scale}} = \begin{cases} \left(\sum_{\mathbf{v}_i \in V^-} g(|d_i|, \mathbf{u}_i) - s\right)^2, & \text{if } N_{\text{cluster}} > 1, \\ 0, & \text{otherwise.} \end{cases} \tag{3}$$

where $g(|d_i|, \mathbf{u}_i) = \frac{||\mathbf{u}_i|| - |d_i|}{||\mathbf{u}_i||}$ defines the target scale to reduce collision regions. $N_{\text{cluster}}$ denotes the number of distinct clusters formed without SDF sign flipping. As shown in Figure 3, two surface points $i$ and $j$ with $d_i \leq 0$ and $d_j \leq 0$ belong to different clusters, thus push the object to be smaller.

**Stability Constraints.** Objects are dragged by gravity and rest on their bottom contacting surface. We ensure stability by enforcing contact between an object's bottom points and its parent surface, where their SDF values should be zero, as shown in Figure 3. The stability loss is defined as:

$$\mathcal{L}_{\text{stability}} = \sum_{\mathbf{v}_i \in V^B} \left(1 - e^{-d_i^2}\right) \tag{4}$$

where $\mathbf{V}^B$ are the sampled points on the bottom surface of the bounding box, and $d_i$ are their corresponding SDF values. Further details on collision loss optimization are provided in the *Appendix*.

## 3.4 SPATIAL COHERENCE JUDGMENT

After pose optimization, we apply a GPT-5–based judge to compare the generated 3D scene with the guidance image and planned objects, verifying inter-object relations. To assess this alignment, we design three metrics: (1) object category accuracy, comparing the generated scene with the image guidance; (2) object orientation alignment, measuring how well object orientations match the reference layout; (3) overall spatial coherence, capturing the holistic consistency of the scene layout. Each metric is normalized between 0 (lowest) and 1 (highest). If any metric falls below a predefined threshold, the scene judge triggers a re-planning step. Further details are provided in the *Appendix*.

## 4 EXPERIMENT

**Implementation.** We use GPT-5 (OpenAI, 2025) as the LLM and image generation in vision module. Following Holodeck (Yang et al., 2024c), we retrieve 3D models from a high-quality Objaverse

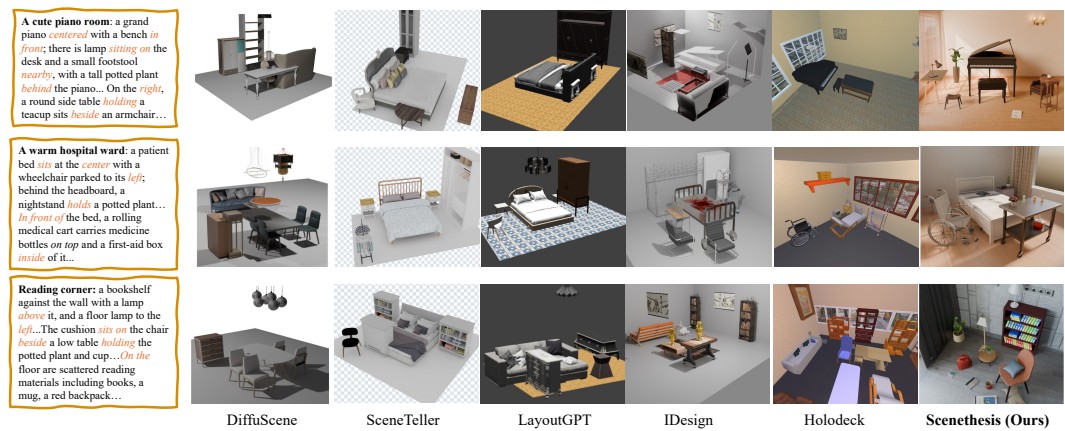

Figure 4: **Qualitative results** of *Scenethesis* compared with text-to-3D scene baselines.

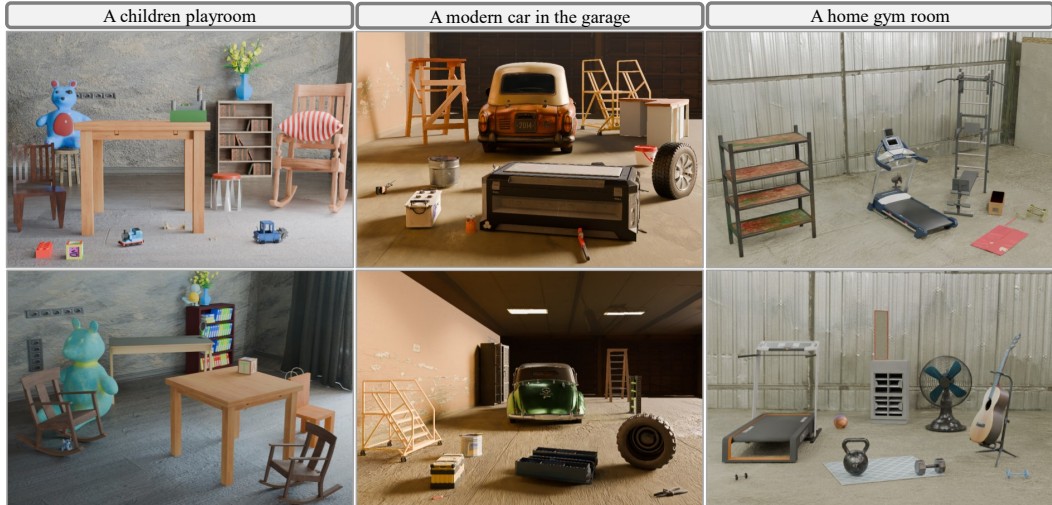

Figure 5: **Output Diversity.** Given the same text prompt, *Scenethesis* can generate diverse scene with various objects and different layouts. More indoor / outdoor examples can be seen in *Appendix*.

subset (Deitke et al., 2023b). Other module details are discussed in the above section. The physics-aware optimization is implemented in PyTorch (Paszke et al., 2019) and PyTorch3D (Ravi et al., 2020). Experiments are run on an A100 (40G) GPU. Further module details appear in the *Appendix*.

**Baselines.** We target interactive scene generation with per-object assets; thus monolithic single-geometry methods are out of scope. For text-to-3D, we evaluate open-source implementations of DiffuScene, LayoutGPT, IDesign, SceneTeller, layoutVLM, and Holodeck (Tang et al., 2024; Feng et al., 2024; Çelen et al., 2024; Öcal et al., 2024; Yang et al., 2024c; Sun et al., 2025). For image-conditioned baselines, we compare to MIDI and Digital Cousins (Huang et al., 2025a; Dai et al., 2024), using the image guidance as the reference. For fairness, all text-to-3D baselines use the same GPT version. We restrict comparisons to methods with publicly available code/models; methods without released code are excluded.

**Setup.** *Scenethesis* generates both indoor and outdoor scenes (Figure 1). As most baselines target *indoor* synthesis, we report comparisons on our indoor subset and present outdoor results separately.

We evaluate diversity and realism on 34 prompts (22 indoor, 12 outdoor) spanning 8 primary and 16 secondary categories from DL3DV-10K (Ling et al., 2024). *Indoor:* **Residential** (living room, playroom, garage, warehouse), **Shopping** (bookstore, store), **Tourism** (museum, piano showroom),

**Sports** (gym, billiards), **Medical** (ward), **Education** (laboratory). *Outdoor:* **Nature/streetscapes** (square, street, beach), **Parks & recreation** (playground).

Since DiffuScene, LayoutGPT, and SceneTeller are designed/trained for residential indoor scenes, we restrict quantitative comparisons to that domain. To mitigate view bias, each scene is rendered from two standardized viewpoints (68 images total). For baselines that omit backgrounds (e.g., SceneTeller), we render *Scenethesis* without environment maps to match their setting.

## 4.1 METRICS

We evaluate four aspects of scene generated methods: *controllability*, *layout realism*, *physical plausibility*, and *interactivity*. Scores are computed per rendered view and averaged per scene.

**Controllability.** We measure prompt adherence for text-based baselines (DiffuScene, SceneTeller, Holodeck, LayoutGPT, IDesign) using: (1) *CLIP Score* (Radford et al., 2021) – cosine similarity between CLIP image and text embeddings. (2) *BLIP Score* (Li et al., 2023b) – image-text matching probability from BLIP-2's ITM head. (3) *VQA Score* (Lin et al., 2024b) – accuracy of answers to prompt-derived questions from a frozen VQA model.

**Layout Realism.** We assess how well scenes reflect real-world layouts using a mix of GPT-5, a human-aligned evaluator in text-to-3D tasks (OpenAI, 2025; Lin et al., 2025b) and human judgments: (1) *Object Diversity* – number of objects and categories in the scene. (2) *Layout Coherence* – adherence of object positions and orientations to common sense. (3) *Spatial Realism* – presence of diverse spatial relations (e.g., *on top of, inside, under*). (4) *Overall Performance* – alignment of object categories and styles with the scene type. Evaluation details and examples are in the *Appendix*.

**Physical Plausibility.** (1) *Col-O* – average object collision rate, (2) *Col-S* – average scene collision rate, (3) *Inst-O* – average object instability rate, and (4) *Inst-S* – average scene instability rate.

Collision is tested via mesh-mesh intersections, while instability follows Atlas3D protocol (Chen et al., 2024b): average pose change after a short quasi-static physics rollout (Macklin, 2022).

**Interactivity.** To ensure objects are accessible and manipulable in the scene based on their functional roles, we follow evaluation metrics from PhyScene (Yang et al., 2024b): (1) *Reach* – average object reachability rate, and (2) *Walk* – ratio of the largest connected walkable area over all walkable regions.

## 4.2 QUANTITATIVE EVALUATION

**Controllability.** Table 1 presents a comprehensive evaluation of text-image alignment. Among all baselines, *Scenethesis* achieves the highest CLIP, BLIP, and VQA scores, confirming its effectiveness in adhering to the text description and the reliability of our agentic pipeline.

**Layout Realism.** Table 1 also shows visual-quality preferences from humans and GPT-5 (OpenAI, 2025; Lin et al., 2025b). Although DiffuScene, MIDI, SceneTeller, and LayoutGPT are trained on 3D-FRONT–style indoor data, the training-free *Scenethesis* matches or exceeds their layout realism on residential scenes. Across broader indoor categories (e.g., shopping, tourism, sports), Table 1 shows that *Scenethesis* significantly outperforms Digital Cousins, Holodeck, and IDesign in visual quality and spatial realism, highlighting the benefit of vision-guided refinement.

**Physical Plausibility and Interactivity.** As shown in Table 2, *Scenethesis* markedly reduces collisions and instability on indoor scenes. Many baselines (DiffuScene, SceneTeller, LayoutGPT, MIDI) do not optimize mesh-level contact or stability during generation, leading to frequent interpenetrations. Although LayoutVLM tries to leverage on the layout common sense knowledge from VLM, the VLM layout guidance is noisy (Sun et al., 2025), still leading to unrealistic layout pattern similar to LLM-based method, e.g. the bookshelf or the chair orients towards the walls (see *Appendix*). Besides, LayoutVLM relies on IoU-based loss to avoid collision. This inaccurate geometry approximation does not model small object stability, leading to floating/unstable small objects standing on top of big furniture. Others (Digital Cousins, IDesign, Holodeck) operate at 3DBBs/collider granularity—with post-hoc cleanups such as wall snapping, $z$ de-penetration, and $x$–$y$ nudging—which remain coarse and yield overlaps and brittle *child-on-parent* placements (small items on furniture). By contrast, *Scenethesis* enforces SDF-based *contact/support* constraints *in-loop*, coupled with semantic pose alignment, driving layouts toward collision-free and support-stable configurations. Beyond physical

Table 2: Physical-plausibility and interactivity results.

| Method | Physical Plausibility | | | | Interactivity | |
|---|---|---|---|---|---|---|
| | Col-O↓ | Col-S↓ | Inst-O↓ | Inst-S↓ | Reach↑ | Walk↑ |
| MIDI | 15.1% | 50% | 51.95% | 90.90% | 0.75 | 0.63 |
| Digital Cousins | 15.2% | 60% | 24.0% | 88.33% | 0.86 | 0.87 |
| DiffuScene | 19.5% | 55% | 20.75% | 83.33% | 0.74 | 0.83 |
| SceneTeller | 35.2% | 75% | 41.17% | 78.57% | 0.75 | 0.80 |
| Holodeck | 6.1% | 21% | 7.00% | 31.58% | 0.90 | **0.96** |
| LayoutGPT | 33.3% | 73% | 51.2% | 81.14% | 0.76 | 0.81 |
| LayoutVLM | 12.2% | 57.1% | 20.3% | 71.4% | 0.90 | 0.71 |
| IDesign | 6.51% | 65% | 8.3% | 68.88% | 0.88 | 0.80 |
| **Ours** | **0.8%** | **6%** | **3.20%** | **16.67%** | **0.94** | **0.96** |

plausibility, *Scenethesis* also achieves higher reachability and walkability, producing more accessible, navigable scenes aligned with object functionality.

**Outdoor scenes.** We evaluate outdoor generations on text–image alignment and physical plausibility (Table 3). We omit preference scores (no baseline explicitly targets outdoor scenes) and Reach/Walk metrics (no area restricted

Table 3: Outdoor Scene Qualitative Evaluation.

| | Text–Image Alignment | | | Physical Plausibility | | | |
|---|---|---|---|---|---|---|---|
| | CLIP↑ | BLIP↑ | VQA↑ | Col-O↓ | Col-S↓ | Inst-O↓ | Inst-S↓ |
| **Ours** | 29.87 | 71.62 | 0.7592 | 0.06% | 3.3% | 0.12% | 6.7% |

floor plan). Performance is comparable to our indoor results (Table 2), indicating robustness. Despite the added difficulty of non-planar terrain—objects must align to uneven surfaces for stability—*Scenethesis* maintains low collisions and strong stability beyond indoor settings.

## 4.3 QUALITATIVE EVALUATION

**Text-to-3D baselines.** Figure 4 shows side-by-side comparisons; image-based baselines are in the *Appendix*. Methods trained on 3D-FRONT–style data (LayoutGPT, SceneTeller, DiffuScene) favor large furniture, often collide, and rarely follow prompts with supporting items or fine-grained relations (*on/inside/sit*). IDesign and Holodeck broaden categories and follow prompts but frequently misorient key objects (e.g., chairs facing walls, shelves blocking the windows) and exhibit inter-object collisions relying on 3DBBs approximation or predefined supports (see additional examples in *Appendix*).

*Scenethesis*. In contrast, *Scenethesis* is able to place small objects *on* or *inside* the supports and maintains collision-free, stable layouts. It also enables fine-grained positioning across support levels (e.g., shelves, carts, as illustrated in Figure 6), which is crucial for embodied-AI scenarios that require meaningful object manipulation (Yang et al., 2024b; Nasiriany et al., 2024). LLM-only planners lack this degree of visual grounding and typically fail on such spatial realism. Figure 5 further highlights diverse outputs from the same prompt (both asset selection and layout).

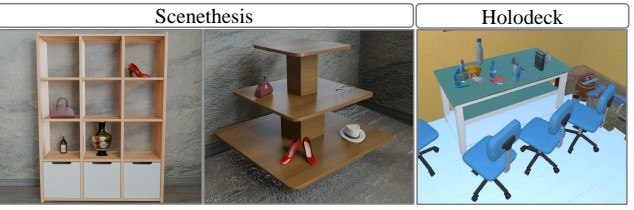

Figure 6: *Scenethesis* precisely places small objects (e.g., bag, wine bottle, shoes, vase) within shelf compartments rather than only on top (Holodeck).

To assess downstream utility, we demonstrate that *Scenethesis* can be used as a data engine for virtual content creation by using *Scenethesis* to improve a feed-forward interactive scene generator in the *Appendix*.

## 4.4 ABLATION STUDY

The physics-aware optimization has three components: *pose alignment*, *collision constraint*, and *stability constraint*. We perform ablation studies to assess their effectiveness.

Table 4: Ablation study on the effectiveness of adding each component in spatial and physical constrains.

| Component | Pose Align. ↑ | Collision ↓ | Instability ↓ |
|---|---|---|---|
| Raw layout | 0.536 | 22.7% | 87.3% |
| +Pose Alignment | 0.732 | 10.6% | 74.2% |
| +Collision | 0.755 | 3.6% | 69.8% |
| +Stability | **0.836** | **0.8%** | **3.2%** |

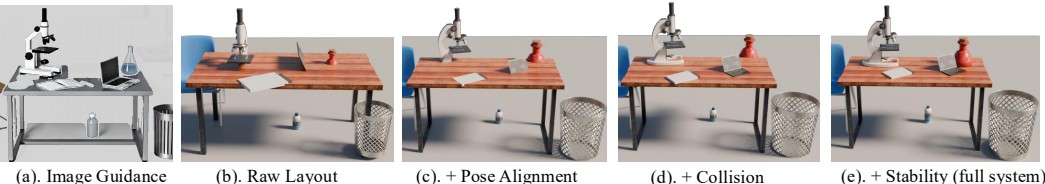

(a). Image Guidance     (b). Raw Layout     (c). + Pose Alignment     (d). + Collision     (e). + Stability (full system)

Figure 7: **Effects of different constraints.** (a) image guidance from text input. (b) *Raw layout*: places 3D models in estimated 3DBBs. (c) + *Pose alignment*: adjusts 5DoF poses to align the pose. (d) + *Collision*: adding collision constraint. (e) + *Stability*: adding stability constraint.

**Metric.** For each generated scene, we render the output and assess *pose alignment* to the image guidance with GPT-5 based on: (1) per-object orientation/scale/position and (2) global spatial coherence. The judge returns a similarity score in [0,1] (higher is better). We also measure mesh collisions and support instability using the protocol in Section 4.2

**Baselines.** *Raw Layout*: Objects are placed on the locations of 3DBB . *Pose Alignment*: Aligns object pose with image guidance via correspondence matching. *Collision Constraint*: Optimizes location / size to avoid collisions. *Stability Constraint*: Optimizes location to remain stable.

**Results.** As shown in Table 4, pose alignment significantly improves spatial consistency, while collision and stability constraints enhance physical plausibility, making scenes simulation-ready. Figure 7 shows qualitative visualization.

## 5 CONCLUSION AND LIMITATION

We introduce *Scenethesis*, a training-free agentic framework for generating high-fidelity interactive 3D scenes by leveraging LLM-based coarse scene planning, vision-guided layout refinement, and physics-aware optimization for object pose adjustment. A scene judge module ensures spatial coherence. Experimental results demonstrate that it significantly outperforms SOTA baselines in layout coherence, spatial realism, and plausibility. Our approach is limited by retrieval databases since generative 3D methods cannot yet handle articulation. Future advances in generative 3D could overcome this constraint by enabling articulated object synthesis, enhancing scene diversity.

## REPRODUCIBILITY STATEMENT

We provide detailed pseudo-codes in the Appendix. In addition, our code, configuration files, and asset lists will be released. **Data/Assets.** We use a curated Objaverse subset and HDR environment maps; the exact lists and licenses will be provided with the release. **Pretrained Models.** Grounded SAM Ren et al. (2024) and DepthPro Yang et al. (2024a) are used off-the-shelf (versions/commits will be documented in the repo). **LLM/VLM.** GPT-5 is used for coarse planning, scene layout judging, and quality evaluation. All used prompts are listed in APP. B.2. **Implementation.** The physics-aware optimization is implemented in PyTorch Paszke et al. (2019) and PyTorch3D Ravi et al. (2020). **Compute.** All experiments ran on a single NVIDIA A100 (40G) GPU.

## ETHICS STATEMENT

Generated scenes are for research/creative use and are not safety-certified. We respect dataset licenses (Objaverse, HDRIs) and use no personal data. GPT-5 is used under human supervision (details in the Appendix).

## ACKNOWLEDGEMENTS

We thanks Xuan Li for his insightful and constructive feedback on our method discussion.

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

## A    LLM Usage

We used **LLM** (OpenAI, 2025) in two ways: (i) *manuscript assistance* for grammar check only—no technical claims, equations, figures, code, or results were authored by the model; and (ii) *in-method* roles as a coarse layout planner. All outputs were reviewed by the authors; humans made final decisions and accept full responsibility.

**Model and settings.**    Unless otherwise stated, we used the `gpt-5` API. Prompts are provided in *Appendix* B.2. No personal or sensitive data were sent to the API.

**Scope limitations.**    The LLM did not select baselines, choose hyperparameters, or modify experimental results/code. Its layout planning is described in Sec. 3 and APP. B.2.1.

## B    Implementation Details of *Scenethesis*

### B.1    Algorithm Overview

In this section, we provide a high-level algorithmic overview of *Scenethesis*, with detailed steps outlined in Algorithm 1.

### B.2    Method Details

#### B.2.1    Coarse Scene Planning

Using the user's scene prompt as input, the LLM (powered by GPT-5 (OpenAI, 2025)) follows a six-step process:

1. Interpreting the user's scene prompt.
2. Reviewing the object categories available in the provided asset database.
3. Selecting relevant objects from the asset list.
4. Cross-checking the availability of the selected objects.
5. Planning the scene using the selected objects.
6. Generating output files according to the specified standards.

The final coarse scene planning output consists of two components: a list of selected object categories commonly found in the scene (defining anchor object and other common objects) and an upsampled prompt that outlines the scene's spatial hierarchy. The designed prompt presents in *Coarse Scene Planning Instruction Prompts* Section D.1 and the output example is in *Coarse Scene Planning Output Example* Section D.2.

#### B.2.2    Layout Visual Refinement

Based on the upsampled prompt, GPT-5 generates an image to serve as fine-grained layout guidance. Several post-processing steps are applied to this image guidance:

- Scene Graph Construction: GPT-5 (OpenAI, 2025) is used to generate a scene graph, defining the ground as the *root object*, along with *parent objects* and their corresponding *child objects*. Additionally, Grounded-SAM (Ren et al., 2024) segments each object in the image to obtain masks and cropped images. These are then projected into 3D space using Depth Pro (Bochkovskii et al., 2024), allowing for the initial positioning of objects within a spatial relationship graph.

- Asset Retrieval. CLIP (ViT-L/14 trianed on LAION-2B) image and semantic features are employed to retrieve 3D assets that align with the image guidance. GPT-5 (OpenAI, 2025) is further utilized to select the most relevant environment map based on the upsampled prompt. It is important to note that *Scenethesis* focuses on layout planning for objects on the ground, while background elements, such as wall decorations, lighting, or outdoor settings (e.g., sunshine or the sea), are visually determined by the environment map.

---

**Algorithm 1** Text to 3D Interactive Scene Generation

---

1: **Input:** User text
2: **Output:** 3D interactive scene layout
3:
    **Stage 1: Coarse Scene Planning :**
4:    object_list, upsampled_prompt ← LLM(user_text)    ▷ obtain the object list and an upsampled prompt
5:
6:
    **Stage 2: Layout Visual Refinement :**
7:    img_guidance ← VLM (upsampled_prompt)    ▷ generate the guidance image as the reference
8:    cropped_images ← Grounded_SAM (img_guidance, object_list)    ▷ identify each object and crop the images
9:    depth_map ← Depth_Pro (img_guidance)    ▷ generate depth map
10:    5DoF_poses ← Extract_Poses(cropped_images, depth_map)    ▷ generate initial 5DoF poses
11:    scene_graph ← VLM (img_guidance, object_list, 5DoF_poses)    ▷ generate scene graph
12:    3D_assets← CLIP (cropped_images, object_list)    ▷ retrieve 3D assets
13:    environment_map ← VLM (upsampled_prompt)    ▷ retrieve environment maps
14:
15:
    **Stage 3: Physics-aware Optimization:**
16:    scene_SDF ← Init_Scene_SDF(anchor_object)    ▷ compute SDF for each object
17:    **for** node in scene_graph.bfs_traverse() **do**    ▷ iterate over all objects
18:        $s, \mathbf{R}, \mathbf{T}$ ← node.pose    ▷ variables to be optimized
19:        parent_SDF ← node.parent.SDF    ▷ obtain parent object's SDF
20:        **for** iteration = 1 to max_iterations **do**
21:            mesh ← Get_Object_Mesh(node)
22:            mesh$^*$ ← Apply_Transform(mesh, $s, \mathbf{R}, \mathbf{T}$)    ▷ coordinate alignment
23:            img_rendered, depth_rendered ← Render(mesh$^*$, camera)    ▷ render RGB and depth
24:            correspondence ← RoMa(img_guidance, img_rendered)    ▷ correspondence matching
25:            mesh_points ← Get_Point_Clouds(depth_rendered, correspondence, camera)
26:            guided_points ← Get_Point_Clouds(depth_map, correspondence, camera)
27:            $L_{pose\_2D}$ ← Dist_2D(correspondence)    ▷ loss computation
28:            $L_{pose\_3D}$ ← Dist_3D(mesh_points, guided_points)
29:            $L_{collision}$ ← Collision(mesh$^*$, scene_SDF)
30:            $L_{stability}$ ← Stability(bottom_points(mesh), parent_SDF)
31:            $loss$ ← $\lambda\mathcal{L}_{pose} + \lambda_{collision}\mathcal{L}_{collision} + \lambda_{stability}\mathcal{L}_{stability}$
32:            $loss$.Backward()
33:            $optimizer$.Step()    ▷ pose optimization
34:            $optimizer$.Zero_Grad()
35:        **end for**
36:        scene_SDF ← Update_Scene_SDF(scene_SDF, node)
37:    **end for**
38:
39:
    **Stage 4: Scene Spatial Coherent Judgment:**
40:    Multi-view images ← Render (optimized_3D_scene)
41:    Qualified ← VLM (Multi-view images)
42:    **if not** qualified **then**
43:        **goto Stage 1**    ▷ re-generate if current optimization fails.
44:    **end if**
45:
46: **Return:** Optimized 3D interactive scene

---

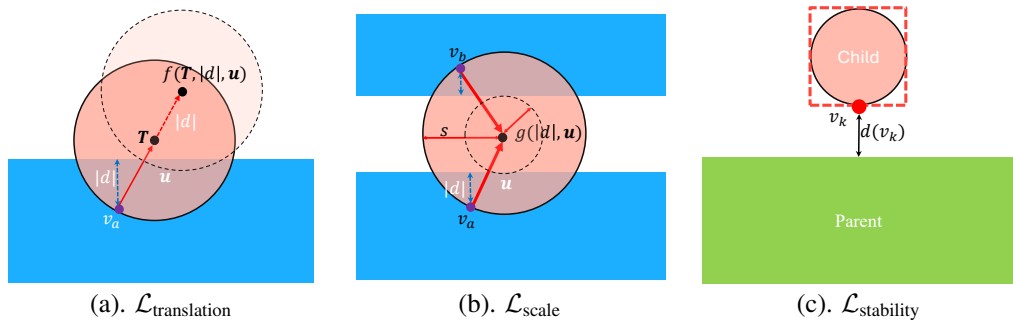

(a). $\mathcal{L}_{\text{translation}}$      (b). $\mathcal{L}_{\text{scale}}$      (c). $\mathcal{L}_{\text{stability}}$

Figure 8: Illustration of *collision avoidance* and *stability maintenance*. The solid-line circle indicates the 3D object's current position, while the dotted-line circle marks its anticipated position. The black dot represents the centroid of the target object, the purple dots indicate surface nodes with negative SDF values, and the red point $v_k$ is the bottom node of the object. (a). The collision pushes the circle object out of the rectangle along the direction from the sampled point to the circle's center by step $|d|$. (b). The collision indicates the object is too large and negative signed distance fields (SDF) points (i.e. point $v_a$ and point $v_b$) are detected from distinct classes of the object during optimization. The collision loss shrinks the object size such that there are no different clusters of negative SDF points on the object surface can be detected. (c). The *stability maintenance* keeps the the child and the parent to be as close as possible with no collision.

The output of the fine-grained layout planning includes the generated image as guidance, a scene graph with the initial poses of the objects, the retrieved assets, and the retrieved environment map. The visual details are presented in the *web*. Note: the algorithm only generates LLM-planned objects on the scene graph instead of generating each object in image guidance—since the image generator may hallucinate extras (e.g. tiny ones in the coffee-table in the *'A living room with reading materials'* at Figure 18). If an LLM-planned category is missed, the self-check agent triggers a second round of planning until it is detectable. This keeps optimization tractable while guaranteeing all planned objects appear in the final scene.

### B.2.3 PHYSICS-AWARE OPTIMIZATION DETAILS

The physics-aware optimization is an iterative optimization process that consists of two key components: **pose alignment** optimization and **physical plausibility** optimization. pose alignment optimization focuses on aligning the position, size, and orientation of 3D models with their counterparts in the image guidance to ensure visual coherence for spatial relationships. Physical plausibility optimization ensures that the 3D models in the scene are free from collisions and maintain stability, contributing to a realistic and physically consistent layout.

**Pose Alignment.** To align the object's position, size, and orientation with its counterpart in image guidance, *Scenethesis* applies the dense semantic correspondence matching from RoMa (Edstedt et al., 2024). Unavoidable discrepancies in texture and shape between image guidance and retrieved assets are mitigated by focusing on high-level semantics over low-level details. That is, minimizing the distance between correspondence points in the rendered image $I$ and the guided image $\tilde{I}$. Suppose there are $N$ objects in the rendered image $I$, each represented by $\mathbf{o}$ and defined by a 5-DoF configuration, which includes scale $s$, upright rotation $\mathbf{R}$, and translation $\mathbf{T} = (t_x, t_y, t_z)$. The counterpart of each object in the generated image $\tilde{I}$ is denoted as $\tilde{\mathbf{o}}$. The objective of ensuring visual coherence is to minimize the distance between corresponding points by optimizing the 5-DoF parameters. This ensures that the spatial positions, size, and orientations of the 3D models are closely aligned with their counterparts in the guided image. The matching process is formalized as:

$$\{p(x,y), \tilde{p}(x,y)\}_i^m = \text{RoMa}(o, \tilde{o}), \tag{5}$$

where $p(x,y), \tilde{p}(x,y)$ are correspondent pair in object $\mathbf{o}$ and $\tilde{o}$. We select $m$ pair points in each optimization iteration with confident score higher than $\tau$. The higher confidence score indicates a higher probability of matching. We minimize the 2D pixel distance and 3D projected point clouds

distance between the matched pair denoted as follows:

$$\mathcal{L}_{pose} = \lambda_{2d}\mathcal{L}_{2d} + \lambda_{3d}\mathcal{L}_{3d}, \tag{6}$$

where $\lambda_{2d}$ and $\lambda_{3d}$ are coefficients of 2D pixel loss and 3D point cloud loss denoted as $\mathcal{L}_{2d}$ and $\mathcal{L}_{3d}$.

**Physical Plausibility.** Physical plausibility ensure generated 3D scenes adhering to fundamental physical principles. Instead of using 3D bounding box (3DBB) as object approximation, *Scenethesis* accurately detects collision state from the surface points of the 3D models using signed distance field (SDF). The *collision avoidance* and *stability maintenance* as illustrated in Figure 8.

The *collision avoidance* affects the translation $T$ by:

$$\mathcal{L}_{\text{translation}} = ||f(\mathbf{T}, |d|, \mathbf{u}) - \mathbf{T}||_2^2, \tag{7}$$

where $f(\mathbf{T}, d, \mathbf{u}) = T + \mathbf{u} \cdot |d|$ computes a collision-free position $\hat{\mathbf{T}}$ by adjusting $\mathbf{T}$ along direction $\mathbf{u}$ with step size $d$. Here, $d$ is the negative SDF value at a collided point $v_i$ such that $d(v_i) \leq 0$ and $|d| = \max(0, -d(v_i))$ is the negative SDF value $d$ after being processed through a ReLU function, meaning only collided points contribute to this collision term. The direction $\mathbf{u}$ is defined from the collision point toward the model's centroid $\mathbf{C}$, guiding objects away from collisions.

The *collision avoidance* affects the scaling $s$ by detecting that object collides from at least two different directions:

$$\mathcal{L}_{\text{scale}} = \begin{cases} \sum_{\mathbf{v}_i \in \mathbf{V}^-} \left( g(|d_i|, \mathbf{u}_i) - s \right)^2 & \text{if } N_{\text{cluster}} > 1 \\ 0 & \text{otherwise} \end{cases}, \tag{8}$$

where $g(|d_i|, \mathbf{u}_i) = \frac{||\mathbf{u}_i|| - |d_i|}{||\mathbf{u}_i||}$ defines the target scale to reduce collision regions. $N_{\text{cluster}}$ denotes the number of distinct clusters formed without SDF sign flipping. As shown in Figure 3, two surface points $i$ and $j$ with $d_i \leq 0$ and $d_j \leq 0$ belong to different clusters, and thus push the object to be smaller.

The *stability maintenance* affects the translation $T$ by:

$$\mathcal{L}_{\text{stability}} = \sum_{\mathbf{v}_i \in \mathbf{V}^B} \left( 1 - \exp(-d_i^2) \right), \tag{9}$$

where $\mathbf{V}^B$ are the sampled points on the bottom surface of bounding box, and $d_i$ are their corresponding SDF values.

**Method Overview** Building on the physics-aware optimization described above, we now integrate pose spatial constraints and physical constraints into the text-to-3D optimization framework. The following function defines the joint optimization of object position, orientation, and scale:

$$\mathcal{L} = \lambda_p \mathcal{L}_{pose} + \lambda_{c\_T} \mathcal{L}_{\text{translation}} + \lambda_{c\_S} \mathcal{L}_{\text{scale}} + \lambda_s \mathcal{L}_{\text{stability}} \tag{10}$$

### B.3 EXPERIMENT DETAILS

**Parameters.** For pose alignment, we select the $m = 100$ correspondence points with matching confidence $\tau \geq 0.6$ in each optimization iteration. Additionally, we uniformly select $n = 400$ samples from the surface of 3D model to accurately detect the collision and stability states in each optimization iteration.

We explored *Adam* and *SGD* as the optimizer during the optimization process. Though *Adam* has been widely applied for training deep neural networks, the adaptive momentum makes the optimization unstable, leading to sub-optimal. So we use *SGD* in our implementation. The optimization implementation is based on pytorch3D (Ravi et al., 2020) and the visualization is rendered using Blender.

We examine *Scenethesis*'s runtime for scene generation using an RTX-4090. The stages include planning (20s), layout optimization (90s per-object), and self-checking (4s), totaling 13 min. For reference, Digital Cousins (another compositional method) takes 18 min for the same scene.

| User specified **long** prompt case | User specified **short** prompt case |
|---|---|
| 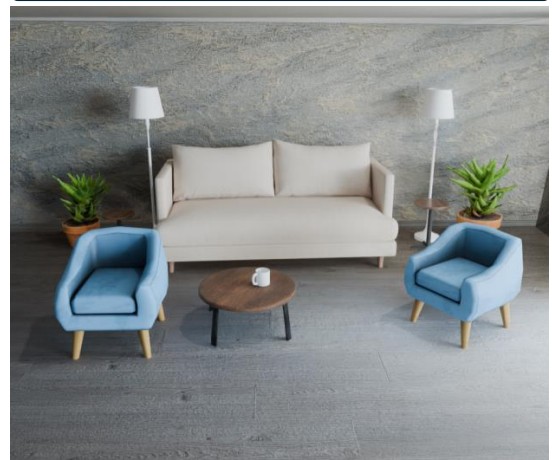 | 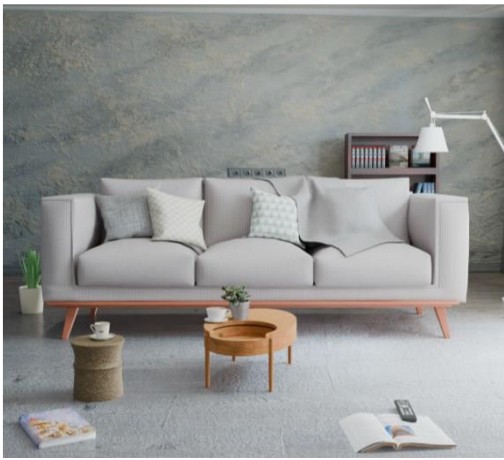 |

Figure 9: *Short prompt*: a living room with reading materials; *detailed long prompt*: A living room that provide a neutral and cozy space with a minimalist design. At the center of the scene, a light beige sofa is positioned against a textured stone wall in the background. In front of the sofa, a round wooden coffee table sits on the floor, with a white coffee cup placed on top. Two blue armchairs are symmetrically arranged on either side of the coffee table, facing inward toward the sofa. Behind each armchair, a tall white floor lamp stands, providing ambient lighting. Next to the lamps, green potted plants are placed near the wall, adding a natural decorative touch.

**Prompts.** *Scenethesis* supports both short and detailed user-specified prompts. A short prompt provides a user-friendly and flexible approach to 3D scene generation, where the LLM interprets the input, revisits the available 3D models in database, selects the common objects and anchor objects, and generates an upsampled text prompt for coarse layout planning. In contrast, a long prompt, which includes user-defined objects and inter-object relationships, enables greater user control over 3D scene generation. In this case, the LLM directly reasons over the detailed prompt, revisits available 3D models in the database, and defines the anchor object, skipping the upsampling stage. We illustrate examples of short and long prompts defining a living room in Figure 9.

**Baseline Comparison.** Our baselines includes: the most relevant text-to-3D scene synthesis methods such as Diffuscene (Tang et al., 2024), SceneTeller (Öcal et al., 2024), LayoutGPT (Feng et al., 2024), IDesign (Çelen et al., 2024), LayoutVLM (Sun et al., 2025), and Holodeck (Yang et al., 2024c); Figure 10 demonstrates the qualitative example of 'a children playroom' for LayoutVLM vs. *Scenethesis*. The results show *Scenethesis* achieves much better layout arrangement, physical plausibility and interactivity comparing with LayoutVLM. We also provide quantitative comparisons with image-conditioned scene methods: Digital Cousins (Dai et al., 2024), an image-based compositional approach that retrieves per-object assets and arranges them based on the image reference, and MIDI (Huang et al., 2025a), an end-to-end diffusion-transformer that jointly reconstructs object meshes and poses from a single image (with instance masks) without retrieval or post-hoc optimization. The quantitative result is presented in the Table 1 and 2 of the main paper.

To ensure a fair comparison with in-context learning based text-to-3D baselines (LayoutGPT, SceneTeller), we adopt a closed-set setting: the retrieval pool is restricted to 3D-FRONT/3D-FUTURE and prompts are limited to living-room and bedroom scenes, matching the baselines' domain coverage. We follow the same evaluation protocol. Comprehensive text–image alignment and visual-coherence results are reported in Table 5, and physical plausibility/interactivity results are in Table 6.

Note, the spatial-quality preference in Table 1 columns report preference for ours over each baseline and a dash in the ours row indicates "not applicable." For example, for 'IDesign', the object diversity metric is 60% / 65%. It indicates in 'object diversity' preference, GPT voted 60% for *Scenethesis* generated scenes (and 40% for IDesign generated scenes) and human voted 65% for *Scenethesis*

Table 5: Quantitative evaluation on text–image alignment and spatial quality (↑ higher is better). Spatial quality preference measures GPT-5 and human preference for **Ours** over baselines.

| Method | Text–Image Alignment | | | Spatial Quality Preference (GPT-5 / Human Evaluation) | | | |
|---|---|---|---|---|---|---|---|
| | CLIP↑ | BLIP↑ | VQA↑ | Object Diversity↑ | Layout Coherence↑ | Spatial Realism↑ | Overall Performance↑ |
| SceneTeller | 26.11 | 54.74 | 0.7801 | 70% / 72% | 80% / 75% | 82% / 75% | 80% / 71% |
| LayoutGPT | 23.01 | 52.15 | 0.7982 | 70% / 65% | 82% / 70% | 90% / 85% | 90% / 75% |
| **Ours** | **32.48** | **82.96** | **0.8342** | – / – | – / – | – / – | – / – |

Table 6: Physical-plausibility and interactivity results.

| Method | Physical Plausibility | | | | Interactivity | |
|---|---|---|---|---|---|---|
| | Col-O↓ | Col-S↓ | Inst-O↓ | Inst-S↓ | Reach↑ | Walk↑ |
| SceneTeller | 33.7% | 70.45% | 43.7% | 72.7% | 0.72 | 0.84 |
| LayoutGPT | 34.2% | 75.0% | 51.2% | 79.6% | 0.74 | 0.80 |
| **Ours** | **0.45%** | **2.27%** | **1.12%** | **9.09%** | **0.95** | **0.96** |

generated scenes (and 35% for IDesign generated scenes). That means both human and GPT prefer *Scenethesis* 's generation results over IDesign's generation results.

For visual quality assessment, we use both a user study and GPT-5 as evaluation tools.

We outline the GPT-5 prompt assessment for both baseline evaluation and ablation evaluation as follows:

- **Comparison with baselines by GPT-5:** GPT-5 is employed to evaluate the generated scenes for four metrics: *object diversity*, *layout coherence*, *spatial realism and complexity*, and *overall performance*. The evaluation prompts are detailed in the *Instruction Prompts for Evaluating Generated Scene* Section D.3. Additionally, a comparison example of generated scenes is provided in Figure 12 with their evaluation results generated by GPT-5 detailed in *Evaluation Example of Generated Scenes* Section D.4.

- **Comparison with baselines by human preference**: We applied a user study to study human preference of baseline method and our method. See Figure 11 as an example. There are 69 users took our survey.

- **Evaluation in ablation studies:** GPT-5 is also utilized to assess the pose alignment metric during the ablation studies of *Scenethesis*'s physics-aware optimization. This evaluation measures the similarity of object position, size, and orientation with their counterparts in the image guidance, as well as the overall spatial coherence of the layout. The instructions for assessing pose alignment are provided in the *Instruction Prompts for Ablation Study* Section D.5.

*We present additional qualitative results.*

- **Qualitative Results of *Scenethesis*'s Scene:** We present different camera views to showcase the qualitative results of *Scenethesis*'s scenes, as shown in Figure 15. Figure 18 presents the generated scenes by *Scenethesis* with their image guidance. Note that *Scenethesis* focuses on layout planning for ground objects. The absence of certain unique assets in Objaverse (Deitke et al., 2023b) may cause discrepancies between the generated scene and the image guidance. Future work could address this by incorporating more diverse assets.

- **Qualitative Results of *Scenethesis* with image-based 3D generation /reconstruction approach.** The quantitative results of image-based methods are presented in the Table 1 and 2 in 4. We provide qualitative comparison results between *Scenethesis* and image-based 3D scene methods in Figure 17. Digital Cousins is an image-based compositional scene synthesis method by object retrieval. In our test results, it often lacks spatial coherence and physical plausibility (e.g., floating items), whereas *Scenethesis* produces collision-free, supported placements. MIDI is a learning-based approach that jointly reconstructs object meshes and poses from a single image(with instance masks). Compared with *Scenethesis*,

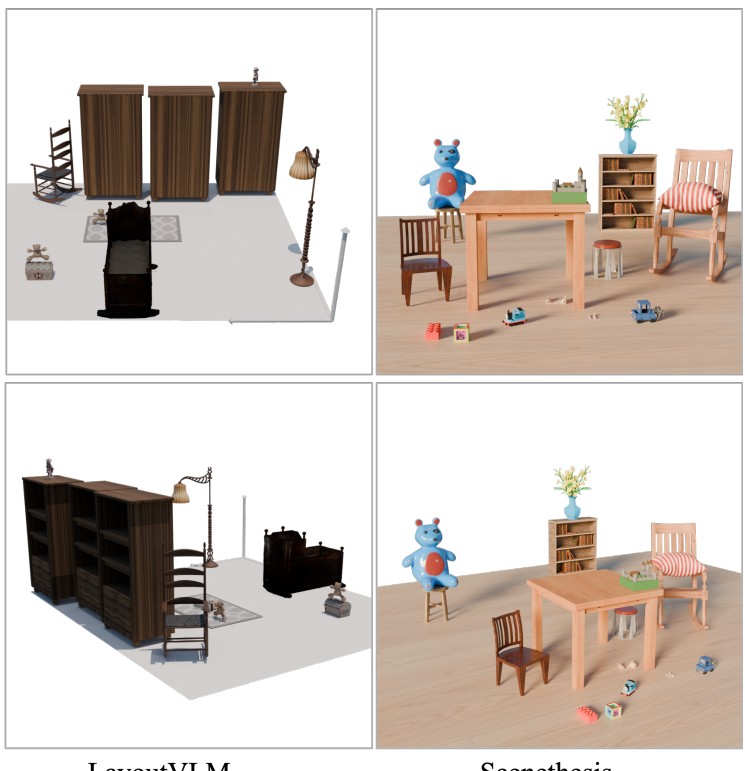

LayoutVLM                    Scenethesis

Figure 10: Qualitative example of children playroom for LayoutVLM vs. *Scenethesis*

we observe: (i) hallucinated geometry under partial views, reducing per-asset fidelity (e.g., artifacts in table, cart, and shelf Figure 17); (ii) errors on long-tail relations—small objects *on/inside/under* larger supports, leading to heavily collision, floating (e.g., the flower vase failing to sit on the shelf Figure 17); and (iii) no explicit physical constraints, leading to floating and penetrations (e.g., shelf, chair, and toys intersect and float Figure 17). These behaviors are consistent with its training prior: layouts learned largely from 3D-FRONT/3D-FUTURE, which emphasize large furniture and underrepresent small-object arrangements. Consequently, MIDI is less simulation-ready. By contrast, *Scenethesis* 's retrieval-based pipeline provides edit-ready, high-quality meshes and, through SDF contact/support constraints, produces diverse, fine-grained relations with physical plausibility. This supports downstream interaction and also serves as a data engine to strengthen future learning-based methods.

- **Qualitative and quantitative Results of Physical plausibility Comparison**: The physical Plausibility quantitative comparison presented in Table 1 4. Among all text-to-3D methods, LayoutGPT, SceneTeller, and Diffuscene only focus on visual quality ignoring physical plausibility. Therefore, we observe high collision rate and instability rate of these methods in the Table 1 and Figure 4 4. While Holodeck and IDesign apply physical constrains via 3DBB approximation. In particular, Holodeck applies both soft and hard constrains based on the Depth-First-Search Solver and small objects are placed on predefined locations. These small objects may collide with each other due to the shape and size variations as shown in Figure 13.

- **Visual Comparison with Holodeck:** In addition to the quantitative comparison presented in Table 1 of the 4, we provide a visual comparison between *Scenethesis* and Holodeck, a state-of-the-art LLM-based 3D interactive scene generation method, in Figure 16. Based on the four evaluation metrics detailed in 4, scenes generated by *Scenethesis* demonstrate greater diversity in object categories, quantities, and sizes. More importantly, *Scenethesis*'s scenes have a broader range of spatial relationships, such as *"on top of"*, *"inside"*, and

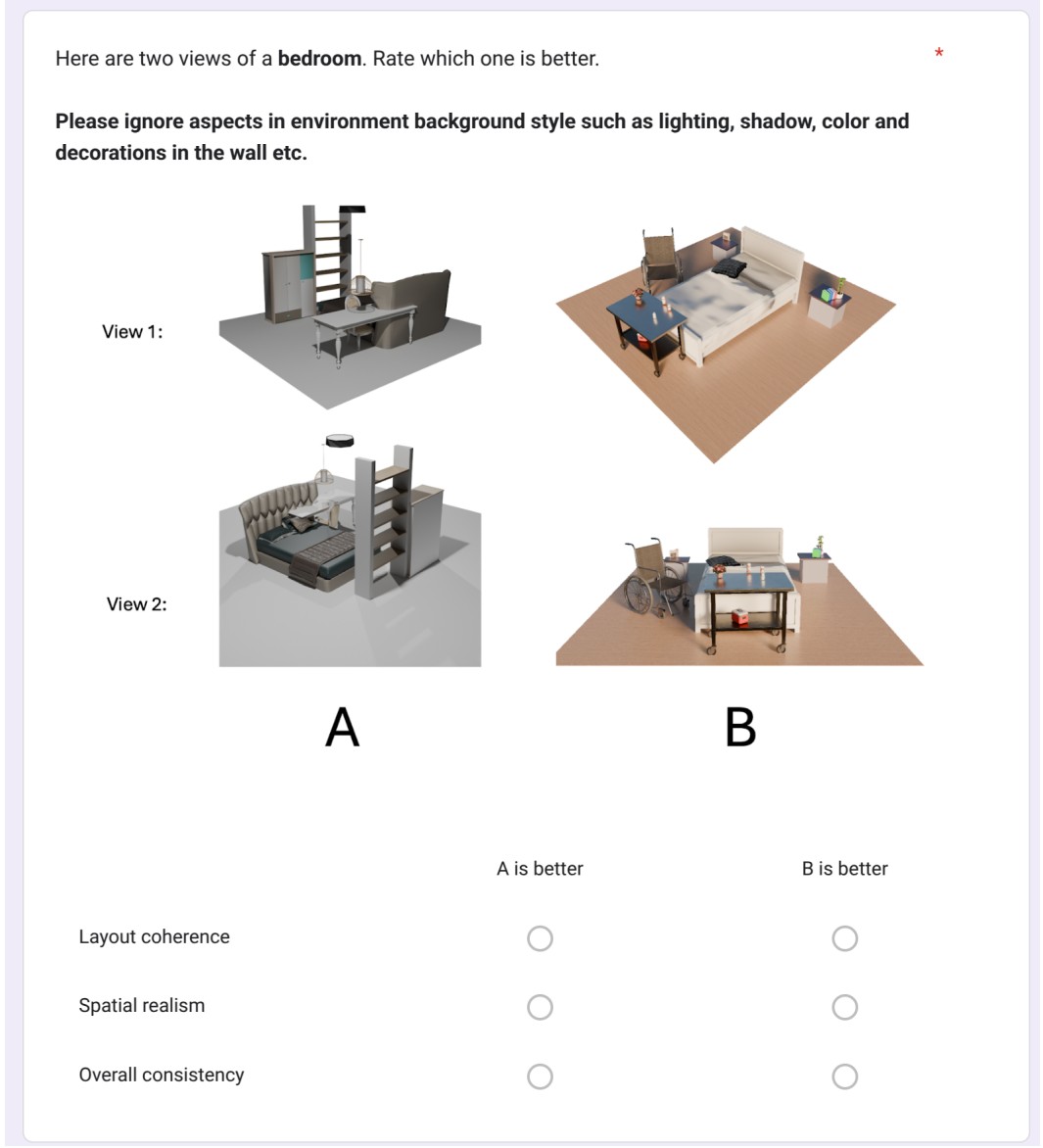

Figure 11: **User study example.**

*"under"*, compared to those generated by Holodeck (Yang et al., 2024c), which supports only *"on top of"* spatial relation. Furthermore, *Scenethesis*'s scenes align more faithfully with the intended scene type. i.e. when given the description *"a peaceful beach during sunset"*, *Scenethesis* produces an outdoor scene with appropriate beach elements, while Holodeck incorporates beach-related objects but generates an environment resembling an indoor setting.

**Success rate.** In our quantitative evaluation, 72% of scenes could pass the judge after the first optimization round; the self-check lifts the success rate to 91% after the first round repair and 2-3 repairs for the rest scenes, confirming both the pipeline's robustness and the value of the self-check agent.

**Limitations.** We observe three main failure modes: (1) LLM-planned objects that are *very small* in the generated image are missed by the detector; (2) *Heavy occlusion* degrades pose alignment. Our pose alignment optimizes scale, translation, and rotation of a *category-matched* mesh using

| Scenethesis | Holodeck |
|---|---|

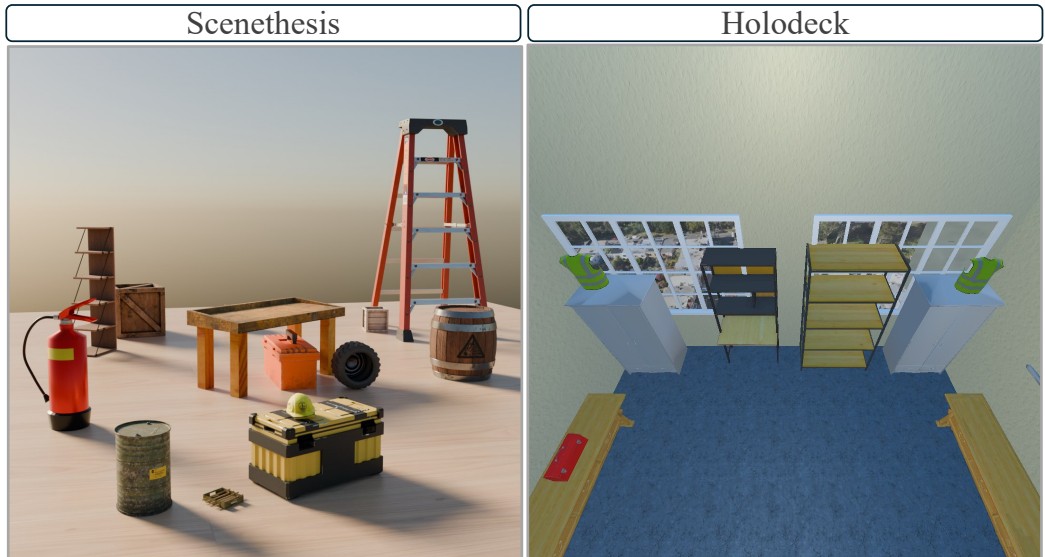

Figure 12: An example comparison of generated scenes given user prompt: "a warehouse". Note that *Scenethesis*'s scenes are rendered without an environment map to ensure a fair comparison with Holodeck's scenes.

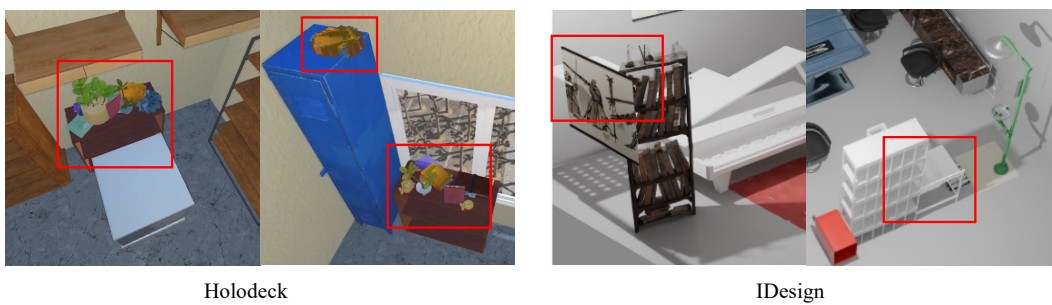

Holodeck               IDesign

Figure 13: An example of objects collision from Holodeck and IDesign.

semantic correspondences computed only on an object's *visible* pixels. The loss in *Eq. 9, Appendix.* are back-propagated to pose until matched 3D points align with the visible region while ensuring physically plausible. This category-level matching tolerates asset discrepancy and works reliably with $\geq 70\%$. For example, partially occluded lamp in the *'A living room with reading materials'* scene in Figure 18). When visibility drops further, correspondences become sparse/noisy and the optimizer can shrink or misplace objects (e.g., the shelf in *"A living room with reading materials"* and the corner sofa in *"a cute piano room"*, Figure 18). Improving correspondence robustness for tiny and heavily occluded objects is an important direction for future work; (3) *Retrieval database.* Our approach is limited by retrieval databases since generative 3D methods cannot yet handle articulation and high-quality assets during occlusion (as see in MIDI's example in Figure 17). Future advances in generative 3D could overcome this constraint by enabling articulated object synthesis, enhancing scene diversity.

## C  DOWNSTREAM APPLICATION

***Scenethesis* as a data engine.**  To assess downstream utility, we demonstrate that *Scenethesis* can be used as a data engine for virtual content creation by using *Scenethesis* to improve a feed-forward interactive scene generator (MIDI-3D). Existing feed-forward interactive scene generation method (e.g., MIDI-3D (Huang et al., 2025a)) are trained on 3D-FRONT, whose supervision is dominated by large furniture and underrepresents common functional relations (e.g., small-on-large support, containment/inside, under). Since *Scenethesis* produces scenes with diverse layouts and rich

| Method (training data) | CD-S ↓ | F-score-S ↑ | CD-O ↓ | F-score-O ↑ | IoU-S ↑ |
|---|---|---|---|---|---|
| MIDI-3D (3D-FRONT) | 0.0416 | 0.6186 | 0.0455 | 0.7466 | 0.5644 |
| MIDI-3D (3D-FRONT+5K) | **0.0252** | **0.7590** | **0.0316** | **0.7922** | **0.6935** |

Table 7: Quantitative results on BlendSwap. CD-S and CD-O refer to scene-level CD and object-level CD respectively; F-score-S and F-score-O refer to scene-level F-score and object-level F-score respectively. Lower CD is better; higher F-score/IoU is better. *5K* refers to 5K *Scenethesis* scenes.

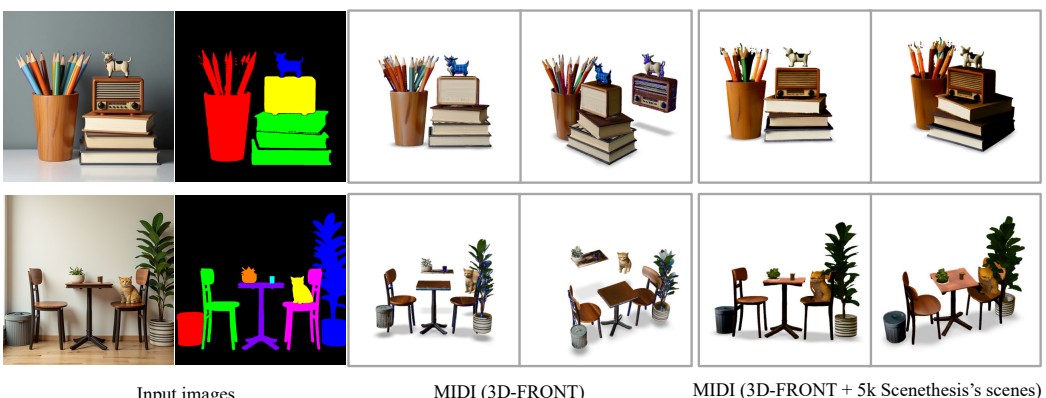

Input images          MIDI (3D-FRONT)          MIDI (3D-FRONT + 5k Scenethesis's scenes)

Figure 14: Qualitative comparison of MIDI generated scenes by training on 3D-FRONT vs. training on 3D-FRONT with additional 5K *Scenethesis* scenes.

spatial relations, we hypothesize that augmenting training with *Scenethesis* scenes will improve the generalization of feed-forward generators.

*Data.* We generate 5k scenes from *Scenethesis* with diverse functional relations (*on/under/inside*).

*Model.* MIDI-3D is trained on 3D-FRONT. As MIDI-3D does not release their training code, we re-implement MIDI by using Trellis (Xiang et al., 2025) as its backbone model.

*Protocol.* We augment the MIDI-3D original training set (3D-FRONT) with 5k *Scenethesis* scenes and using the same data preparation protocol and training hyperparameters.

*Evaluation.* On MIDI-3D's out-of-distribution testing dataset BlendSwap, we report scene- and instance-level Chamfer Distance (CD↓) and F-score (↑, $\tau$=0.1), and layout accuracy via volumetric IoU (↑) between predicted and ground-truth 3D boxes.

*Results.* As summarized in Table 7 and Figure 14, *Scenethesis* augmentation yields consistent gains in geometric quality (lower CD, higher F-score) and layout accuracy (higher volumetric IoU), demonstrating that physically plausible, relation-rich layouts produced by *Scenethesis* provide effective supervision that improves generalization of feed-forward scene generators.

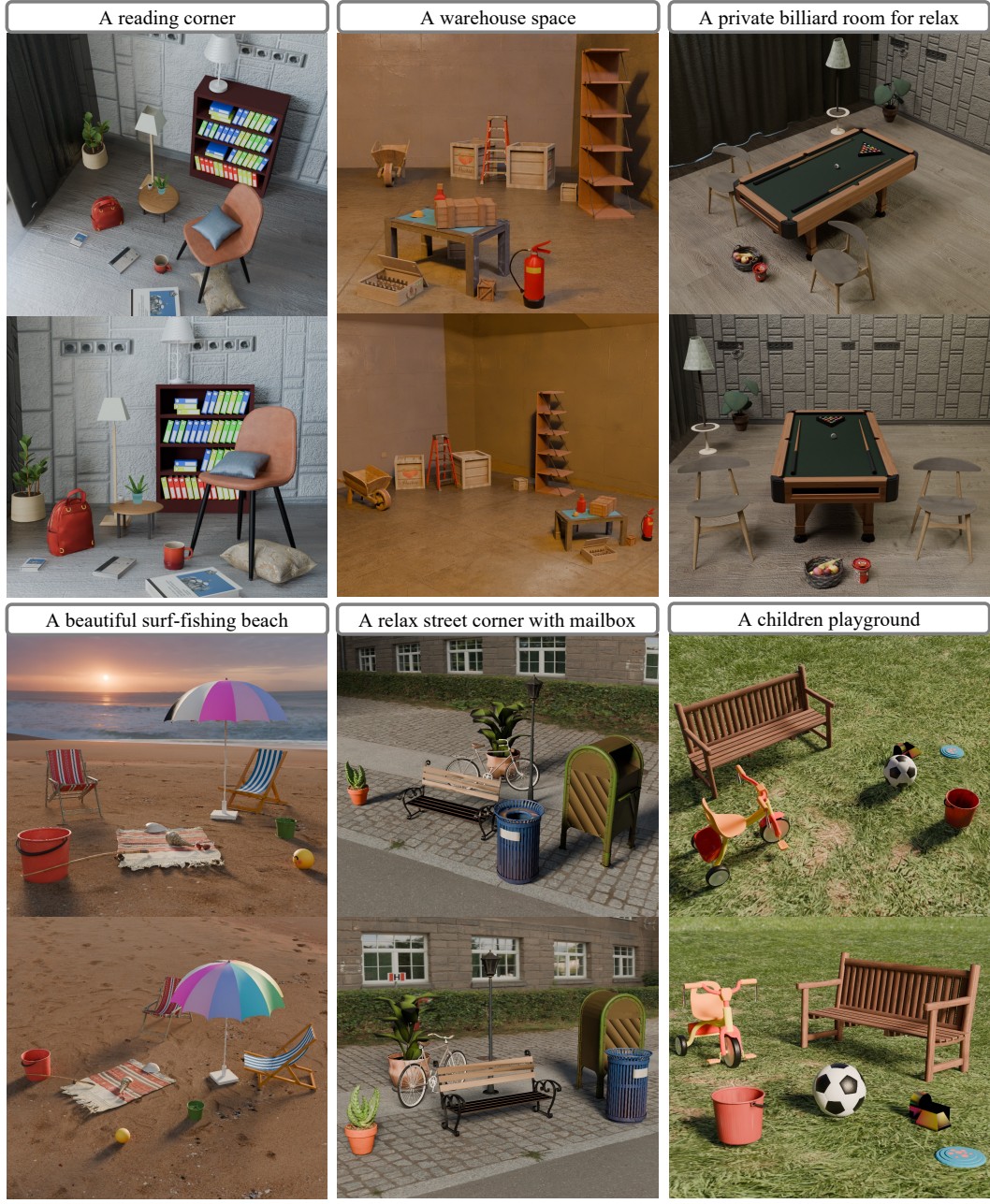

Figure 15: Qualitative results of generated indoor and outdoor scenes by *Scenethesis* at different camera viewpoints

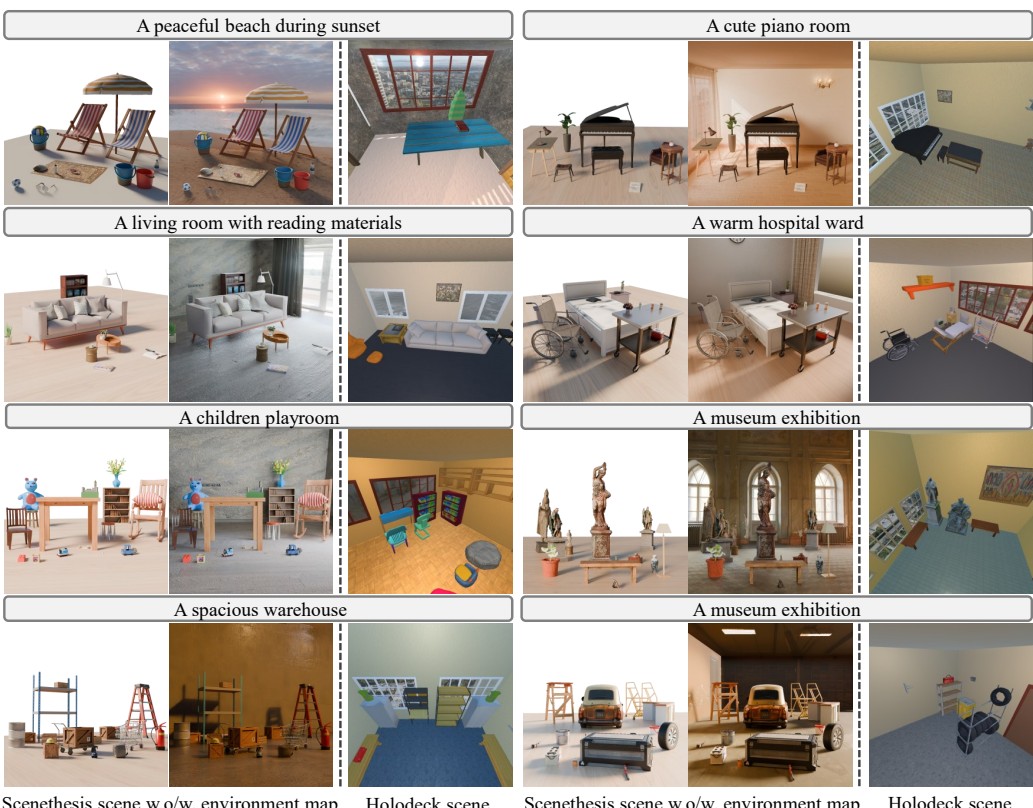

Figure 16: Visualization comparison of generated scenes between *Scenethesis* and Holodeck. The first column of images shows scenes generated by *Scenethesis* without an environment map, the second column displays scenes generated by *Scenethesis* with an environment map, and the third column presents scenes generated by Holodeck. The evaluation metrics, including *object diversity*, *layout coherence*, *spatial realism*, and *overall performance*, are detailed in the **Experiment** section. *Scenethesis*'s scenes have a wider variety of spatial relationships, such as *"on top of"*, *"inside"*, and *"under"*, compared to those generated by Holodeck (Yang et al., 2024c), which supports only *"on top of"* spatial relation. In addition, Holodeck lacks visual perception and usually generates misoriented objects, e.g. shelves occlude the window in children playroom and warehouse, chair orients towards the window in the hospital case, hindering their functionalities.

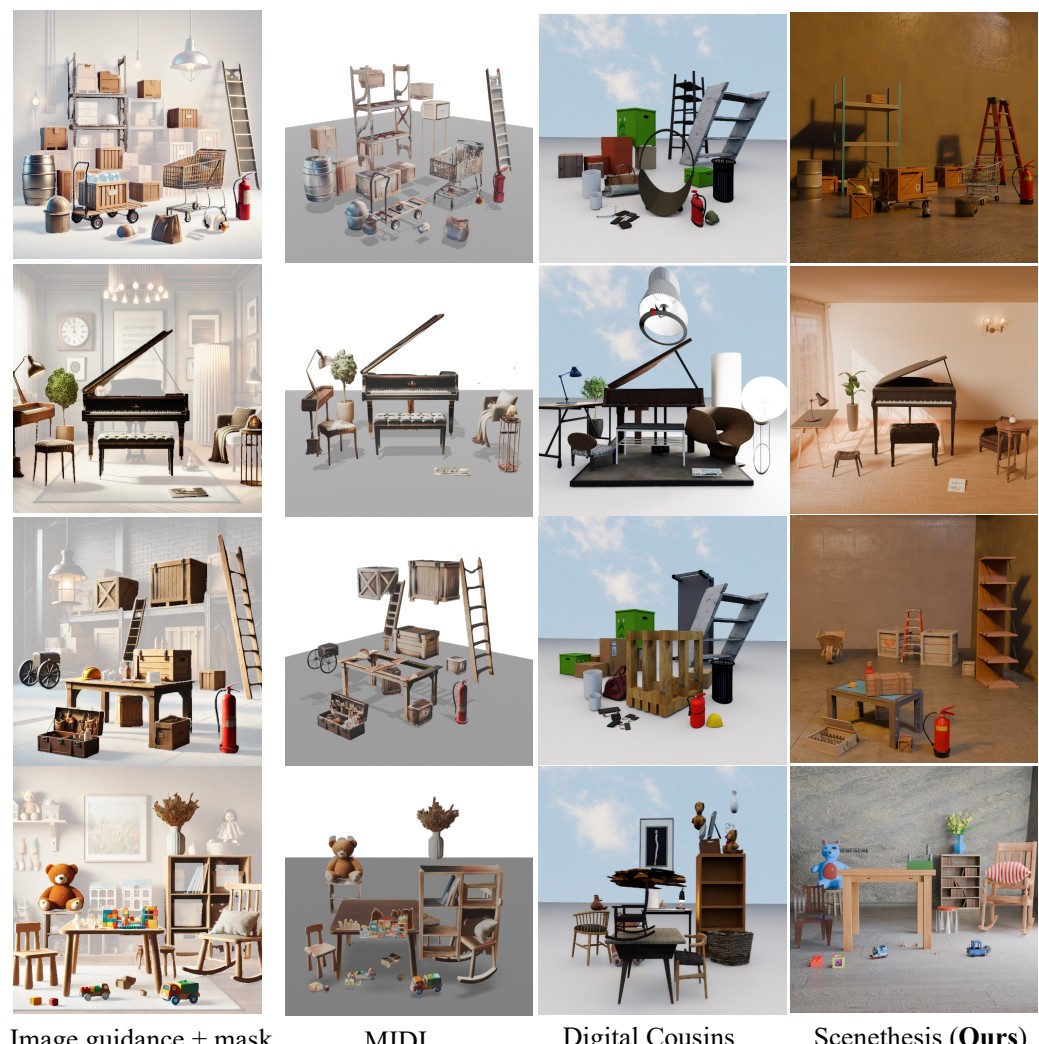

| Image guidance + mask | MIDI | Digital Cousins | Scenethesis (**Ours**) |

Figure 17: Qualitative comparison between *Scenethesis* and single image based 3D scene generation (Digital cousins) /reconstruction methods (MIDI). Digital Cousins is an image-based compositional scene synthesis method by object retrieval. It often lacks spatial coherence and physical plausibility (e.g., floating items), whereas *Scenethesis* produces collision-free, supported placements. MIDI is a learning-based approach that jointly reconstructs object meshes and poses from a single image(with instance masks). Compared with *Scenethesis*, we observe: (i) hallucinated geometry under partial views, reducing per-asset fidelity (e.g., artifacts in table, cart, and shelf Figure 17); (ii) errors on long-tail relations—small objects *on/inside/under* larger supports, leading to heavily collision, floating (e.g., the flower vase failing to sit on the shelf Figure 17); and (iii) no explicit physical constraints, leading to floating and penetrations (e.g., shelf, chair, and toys intersect and float Figure 17). These behaviors are consistent with its training prior: layouts learned largely from 3D-FRONT/3D-FUTURE, which emphasize large furniture and underrepresent small-object arrangements. Consequently, MIDI is less simulation-ready. By contrast, *Scenethesis* 's retrieval-based pipeline provides edit-ready, high-quality meshes and, through SDF contact/support constraints, produces diverse, fine-grained relations with physical plausibility. This supports downstream interaction and also serves as a data engine to strengthen future learning-based methods.

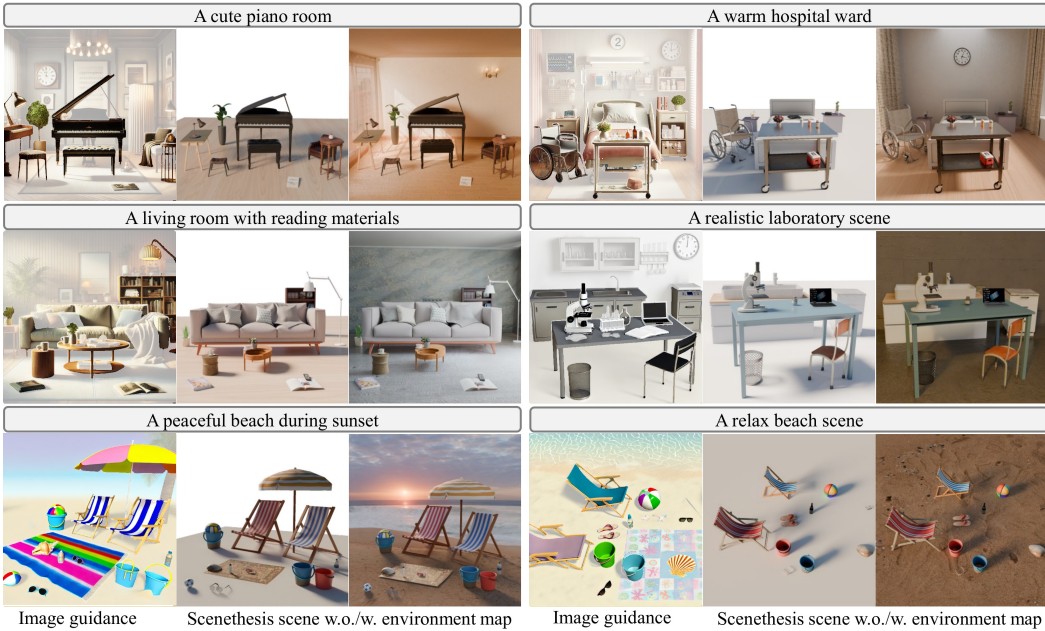

Image guidance    Scenethesis scene w.o./w. environment map    Image guidance    Scenethesis scene w.o./w. environment map

Figure 18: We provide a visual illustration of the generated scenes and their corresponding image guidance. The first column displays the image guidance, while the second and third columns show the generated scenes without and with the environment map, respectively. Note that *Scenethesis* focuses on layout planning for floor-supported objects. *Scenethesis* instantiates only the categories specified by the LLM-planned scene graph, rather than every object hinted by the image guidance—since the image generator may hallucinate extras (e.g., tiny props on a coffee table for *'a living room with reading materials'*). If any planned category is missing from the visual refinement outputs, the self-check agent triggers another round of planning until the object becomes detectable. This design keeps the optimization tractable while ensuring that all planned objects appear in the final scene. Additionally, certain unique assets, such as a beach mat, are unavailable in Objaverse (Deitke et al., 2023b), which may result in the generated scene differing from the image guidance. Future work could enhance the system by incorporating a wider range of assets.

## D PROMPTS EXAMPLES

### D.1 COARSE SCENE PLANNING INSTRUCTION PROMPTS

---

**Coarse Scene Planning Instruction Prompts**

**Task Description:**
You are responsible for generating a set of common objects and planning a scene based on these common objects. You will be given a list that includes all available object categories and a text prompt to describe a scene. This is a hard task, please think deeply and write down your analysis in following steps:

**Step 1: Review All Categories**

    a. Begin by thoroughly reviewing the categories in the provided list.

    b. Identify potential groups or clusters of objects within this list that are commonly found in similar environments (e.g., furniture, electronics, household items, etc.).

**Step 2: Interpret Input Prompt**

    a. Carefully read the input prompt. Understand the theme, primary activities, or the setting it describes, as these will guide your object selection. i.e. if the prompt gives: *children playing room*, then you may think of objects like tent, toy, bear, ball, chair, etc.

**Step 3: Object Selection**

    a. Based on the description, select at least 15 object categories from the list that match the scene.

    b. Determine the anchor object:

        i. Identify the anchor object among the selected objects. Consider the following factors:

            1. A large object directly on the ground (i.e. floor, table, or shelf).

            2. An object that influences where other objects are placed (i.e. a table in a dining room, and there are cups and fruits on the table).

            3. The object should logically anchor the scene and often defines the scene's layout orientation. i.e. the sofa in a front-facing view in the scene.

**Step 4: Object Cross-check**

    a. I will give you $100 tips if you can cross-check whether objects in the scene can be found in the given category list or its relevant categories. i.e., if there is a bookshelf in your planned scene, the bookshelf should also be found in the given list, or bookcase can be found in the list if bookshelf is not covered by the category. Otherwise, re-plan the scene.

**Step 5: Plan Scene with Selected Objects**

    a. Based on the description and selected objects, plan the scene, keeping these aspects in mind:

        i. **Functionality**: Choose objects that are contextually relevant to the scene (e.g., selecting a table, chair, flower vase, and utensils for a dining room), but do not generate any wall décor objects.

        ii. **Spatial Hierarchy**:

            1. Please have a depth effect in the layout. For the depth effect, the scene should have some objects placed on the ground as the background, central, and in the front, resulting in a depth layout. i.e. the sofa and bookshelf are the background of the table and chair set in the living room.

            2. Please have a supportive item in the layout. i.e. the shoes, bag, and hat are in the display shelf in a clothes store, where the display shelf is a supportive item.

        iii. **Balance**: Ensure a mix of large and small objects to avoid overcrowding or under-populating the scene. i.e. taking the table as the center, there are flower vases, fruits, and cups on the table, and chairs are on the sides.

**Step 6: Output Format:**

    a. Save the selected objects as a json file follow the output format:

        **Anchor object**:

        **Other common objects**:

    b. Save scene planning as txt file.

---

D.2  COARSE SCENE PLANNING OUTPUT EXAMPLE

---

**Coarse Scene Planning Output Example**

**User input: a gym room**

**Step 1:** **Reviewing Object Categories for Gym-Related Items**
From the provided list, here are the relevant categories typically found in a gym setting:

- Exercise equipment: dumbbell, treadmill, weight, barbell
- Furniture or storage items: bench, shelf, cabinet
- Accessories and smaller items: towel, water bottle, yoga mat, scale, bag

**Step 2:** **Scene Interpretation**
The scene is a gym room, so the primary objects would focus on equipment for strength and cardio exercises, along with supportive items like shelves for organization and accessories like towels and water bottles for hydration.

**Step 3:** **Scene Layout and Object Selection**

- Anchor object: Treadmill (as the main focus of cardio equipment)
- Other common objects: Dumbbell, Barbell,Bench, Shelf, Water bottle, Towel, Weight, Bag, Mat, Scale, Cabinet, Exercise ball, Fan, Mirror

**Step 4:** **Object cross-check**

- Treadmill - ✓ Available as "treadmill"
- Dumbbell - ✓ Available as "dumbbell"
- Barbell – ✓ Available as "barbell"
- Bench - ✓ Available as "bench"
- Shelf - ✓ Available as "shelf"
- Water bottle – ✓ Available as "water bottle"
- Towel – ✓ Available as "towel rack"
- Weight – ✓ Available as "weight"
- Bag – ✓ Available as "bag"
- Mat – Not explicitly listed, but a close alternative like "place mat"
- Scale – ✓ Available as "scale (measuring instrument)"
- Cabinet – ✓ Available as "cabinet"
- Exercise ball – Not explicitly listed, but a close alternative like by "ball"
- Fan – ✓ Available as "fan"
- Mirror – ✓ Available as "mirror"

**Step 5:** **Plan Scene with Selected Objects**

- Background: Shelf and cabinet are placed to the side or back corner of the room for organization, holding smaller items. Mirror mounted on the back wall for checking exercise form. They serves as supportive equipments...It enhances depth and makes the room feel more spacious...
- Middle Ground: Treadmill is the primary cardio equipment and anchor object, anchoring the scene. Other objects such as bench and barbell are placed around the treadmill, allowing access from multiple directions for various exercises. They provide visual weight to the scene and represent strength-focused equipment...
- Foreground: Dumbbell, water bottle and other objects set closer to the viewer, easily accessible for hand-held exercises. It introduces a smaller, more detailed object in the front. They placed nearby, adding another layer of exercise options for core stability...

This hierarchy ensures a balanced layout, where larger and bulkier items form the background... The middle ground includes the treadmill and training equipment, while smaller and accessible items like dumbbells occupy the foreground... This setup provides a clear layout for practical use and visual depth in the gym scene...

**Step 6:** **Output Format** A json file that include the available objects and a txt file that describe the scene planning.

---

## D.3 Instruction for Evaluating Generated Scenes

---

**GPT-5 Evaluation Prompts for Generated Scenes**

**Evaluate Generated Scenes for *Scenethesis* and Holodeck**
This is a hard problem. You are supposed to compare the alignment of a pair of images with a given text prompt that describe the scene. Images contain generated scenes by two different methods. Please evaluate them in the following five aspects:

1. **Object diversity**: Counting number of object and object types in the scene. The higher number, the better object diversity. i.e. if there are three shelves and a box in the scene. Then the object type is 2 and the number of counting is 3+1 = 4.

2. **Layout coherence**: whether the objects position and orientation in the scene are realistic and adhere to common sense.
   - An ideal layout would be dependents on the scene type. i.e. the objects in the garage scene will forms as a bit chaotic organization and it would be less reasonable if all objects standing against the wall in a very clean order.
   - Objects should be placed reasonably. i.e. shelves fly in the air or hange on the wall.

3. **Spatial realism & complexity**: whether scene contains diverse hierarchy. The measure of spatial complexity is by review scenes and carefully evaluate objects relations. The higher diverse relation indicate a better spatial complexity. The spatial hierarchy refers to the following aspects:
   - relations such as on the top of, in side of, under the, etc.
   - objects in various sizes. i.e. the scene should contain large objects such as shelf and small objects such as apple.
   - objects that are arranged in different area and form a depth effects. i.e. objects are in the background, in the front, and in the center of the space.

4. **Overall performance**: assets' categorizes and styles are faithful to the scene type. i.e. fruits are less likely in the warehouse, while carts, boxes are more likely in a warehouse scene. Thus, it would be a bad performance if find fruits in the warehouse and good performance if carts/shelf/boxes presents in the warehouse. Please carefully evaluate the object's category and scene type. To see whether it is reasonable enough based on your common sense.

5. Please ignore aspects in : environment background style such as lighting, shadow, color, and decorations in the wall etc. This is a challenging task, please think aloud and make your decisions by:
   - analyzing each criteria one by one for each image.
   - report your evaluation results based on the analysis.

6. Your *output format* should be: {} is better than {} in the criteria {}.

---

## D.4 EVALUATION EXAMPLE OF GENERATED SCENES

---

**GPT-5 Evaluation Results**

**Scenethesis's Scene**

1. **Object diversity**:
   - Objects visible: Ladder, barrel, boxes, fire extinguisher, table, tire, crate.
   - Object types: 8 types. Total count: 13
   - Higher diversity due to various objects.

2. **Layout coherence**:
   - Objects are dispersed naturally, suggesting activity.
   - Different sizes and a less structured arrangement suggest a more realistic warehouse scene.

3. **Spatial realism & complexity**:
   - Objects have various spatial relations; ladder stands, boxes on ground, tables at center...
   - Good diversity with a mix of background and central objects.

4. **Overall performance**:
   - Varied objects like ladders, barrels, crates, and extinguishers are typical in a warehouse.
   - Realistic depiction of a warehouse setting.

**Holodeck's Scene**

1. **Object diversity**:
   - Objects visible: Shelves, cabinets, safety vests, and workbenches.
   - Object types: Shelves, cabinets, vests, workbenches. Total count: 7+2=9 (shelves are considered per structure, as are others).
   - Diversity is somewhat low as there are few types.

2. **Layout coherence**:
   - Organization is neat; workbenches and cabinets are along the walls.
   - Reasonable layout for a tidy scene, but lacks variety in sizes.

3. **Spatial realism & complexity**:
   - Shelving and cabinets against walls create vertical space; workbenches on the ground.
   - Lacks diverse spatial relations.

4. **Overall performance**: Objects like shelves and workbenches fit a warehouse setting, but it lacks variety typical in larger-scale warehousing.

**Evaluation**
**Object diversity**: *Scenethesis* is better than Holodeck; **Layout coherent**: *Scenethesis* is better than Holodeck; **Spatial realism & complexity**: *Scenethesis* is better than Holodeck; **Overall performance**: *Scenethesis* is better than Holodeck.

---

## D.5 Instruction Prompts for Ablation Study

---

**Pose Alignment Evaluation Instruction Prompt**

This task involves evaluating the pose alignment between two images in a pair. One image serves as the image guidance (GT), while the other is a generated image. Your objective is to measure the pose alignment of the generated image relative to the GT image. Follow these steps for evaluation:

1. **Review Objects in the GT Image**: Examine all objects in the GT image, focusing on their locations, sizes, and orientations. Understand the spatial relationships among objects, such as *on top of*, *inside*, *under*, etc.

2. **Evaluate pose alignment**: Assess the similarity between the generated image and the GT image based on the following three aspects:

   - Location and Size Similarity: Compare the location and size of objects in the generated image with those in the GT image. Assign a similarity score between 0 and 1, where 1 indicates the highest similarity. For example:
     - If an apple in the GT image is placed at the center of a table, and in the generated image it is placed on the left side of the table, the similarity might be moderate (e.g., 0.5).
     - If the apple is misplaced (e.g., on the ground or missing entirely), the similarity would be very low (e.g., 0.1).
   - Orientation Similarity: Examine the orientation of each object in the generated image compared to the GT image. Pay close attention to details, noting any deviations such as slight tilts (e.g., right/left, up/down) or rotations that create different perspectives. Assign a score from 0 to 1, where 1 indicates perfect alignment and 0 indicates a significant mismatch (e.g., opposite orientation).
   - Overall Layout Similarity: Assess the overall visual coherence of the generated image compared to the GT image, including spatial relationships and hierarchical structure. Assign a similarity score between 0 and 1, where 1 represents a perfect match. For instance:
     - A perfect match occurs when the generated image maintains the same spatial relationships, relative locations, sizes, and orientations as the GT image (e.g., an apple placed at the center of a table in both images).
     - Small deviations in placement or orientation are acceptable but should result in a lower score.

3. **Exclusions**: Do not consider style, appearance, object shape, or texture in your evaluation. Focus solely on pose alignment.

4. **Output Format**: Clearly document your similarity scores for each aspect (Location and Size Similarity, Orientation Similarity, and Overall Layout Similarity) following the format: location and size similarity score is {}, orientation similarity score is {}, and overall layout similarity score is {}. Please save the evaluated scores as a json file.

---

