# OpenReview forum: "Scenethesis: A Language and Vision Agentic Framework for 3D Scene Generation"
_ICLR.cc/2026/Conference — ICLR 2026 Poster_

### Official Review · Reviewer_NE9e · 2025-10-31

**Soundness:** 3
**Presentation:** 3
**Contribution:** 3
**Rating:** 6
**Confidence:** 4

**Summary:**

The paper introduces Scenethesis, a framework that couples LLM-based scene planning with vision-guided spatial refinement and physics-aware optimization for the task of 3D scene generation. Its physics-aware optimization for mesh-level contact and stability improves physical plausibility.

Scenethesis first drafts a coarse layout via an LLM; a vision module then produces a guidance image, constructs a scene graph with 3D bounding boxes, and retrieves assets. A physics-aware optimization module refines object poses using semantic correspondences and SDF-based contact/support constraints, and a final scene judge verifies spatial consistency.

**Strengths:**

1) The pipeline couples LLM-based scene planning with vision-guided spatial refinement to capture real-world spatial complexity.
2) The paper introduces a physics-based optimization that further adjusts asset placement via semantic feature matching and SDF-constraints, which is novel and completes the full scene-generation cycle.
3) The quantitative and qualitative results are strong.
4) The paper is written in an easy-to-follow manner.

**Weaknesses:**

1) Missing results: Table 1 lacks the spatial quality preference results for Scenethesis.
2) Comparison with baselines:

     a. L395-397: SceneTeller and LayoutGPT are training-free in-context learning methods.

     b. Comparing against these methods on 3D-FRONT, which covers primarily bedrooms and living rooms with corresponding assets, is not fair, since many prompt objects are absent. In these baselines, retrieval is typically constrained by predicted category and then nearest dimensions (3DBBs). It is unclear how missing objects in 3D-FRONT are handled in Table 1’s qualitative examples (e.g., retrieving a “double bed” in place of a “piano,” or what substitutes for a “wheelchair”).

     c. To fairly compare with training-free in-context learning methods, the baselines should also be evaluated using Objaverse. As not having the right assets could significantly affect metrics like CLIP score. A small set of sample layouts can be curated using Objaverse and provided to the baselines as in-context samples during evaluation.

3) Lack of comparison with recent baselines (e.g., LayoutVLM [1])

[1] Sun, F. Y., Liu, W., Gu, S., Lim, D., Bhat, G., Tombari, F., ... & Wu, J. (2025). Layoutvlm: Differentiable optimization of 3d layout via vision-language models. In Proceedings of the Computer Vision and Pattern Recognition Conference (pp. 29469-29478).

**Questions:**

All of my questions are listed in the weaknesses section, and I may adjust the rating if they are well addressed.

---

> ### Author Response · Authors · 2025-11-23
>
> ### **Weakness1-Missing results**
> The spatial-quality preference in Table 1 columns report preference for **Ours over each baseline**, so the values appear on the baseline rows by design (see the table caption). A dash in the **Ours** row indicates “not applicable.” For example, for 'IDesign’, the object diversity metric is 60% / 65%. It indicates in ‘object diversity’ preference, GPT voted 60% for _Scenethesis_ generated scenes (and 40% for IDesign generated scenes) and human voted 65% for _Scenethesis_ generated scenes (and 35% for IDesign generated scenes). That means both human and GPT prefer _Scenethesis_ generated results over IDesign generated results. For clarity, we add the explanation in Appendix L1048-1052.
>
> ### **Weakness2-Comparison with text-to-3D scene baselines using closed-set database**
>
> We thanks reviewer's suggestion. We applied additional experiment in the rebuttal to address this concern. To ensure a fair comparison with in-context learning based text-to-3D baselines (such as LayoutGPT, SceneTeller), we adopt a closed-set setting for _Scenethesis_: the retrieval pool is restricted to 3D-FRONT/3D-FUTURE and prompts are limited to living-room and bedroom scenes, matching the baselines’ domain coverage. We use the same rendering setting and follow the same evaluation protocol as described in the paper. Comprehensive text–image alignment and visual-coherence results are reported in Appendix Tab. 7, and physical plausibility/interactivity results are in Appendix Tab. 8.
>
>
> **Table 7. Quantitative evaluation on text–image alignment and spatial quality (↑ higher is better).**
> *Spatial quality preference measures GPT-5 / human preference for **Ours** over baselines.*
>
> | Method      | **CLIP ↑** | **BLIP ↑** | **VQA ↑** | **Object Diversity ↑** | **Layout Coherence ↑** | **Spatial Realism ↑** | **Overall Performance ↑** |
> |-------------|:----------:|:----------:|:---------:|:----------------------:|:----------------------:|:---------------------:|:--------------------------:|
> | SceneTeller |   26.11    |   54.74    |  0.7801   |      70% / 72%         |      80% / 75%         |      82% / 75%        |        80% / 71%           |
> | LayoutGPT   |   23.01    |   52.15    |  0.7982   |      70% / 65%         |      82% / 70%         |      90% / 85%        |        90% / 75%           |
> | **Ours**    | **32.48**  | **82.96**  | **0.8342**|           – / –         |          – / –         |         – / –         |           – / –            |
>
> **Table 8. Physical-plausibility and interactivity results.**
>
> | Method      | **Col-O ↓** | **Col-S ↓** | **Inst-O ↓** | **Inst-S ↓** | **Reach ↑** | **Walk ↑** |
> |-------------|:-----------:|:-----------:|:------------:|:------------:|:-----------:|:----------:|
> | SceneTeller |    33.7%    |   70.45%    |    43.7%     |    72.72%    |    0.72     |   0.84     |
> | LayoutGPT   |    34.2%    |    75%      |   51.2%      |    79.55%    |    0.74     |   0.80     |
> | **Ours**    | **0.45%**   | **2.27%**   | **1.12%**    | **9.09%**    | **0.95**    | **0.96**   |
>
> We add this experiment in Appendix L1042-1046,  Tab.7, and Tab. 8.   The comparison results align with our assessment in Tab. 1 and Tab. 2. _Scenethesis_ outperforms both baselines in their own domain, confirming that improvements stem from layout/physics, not asset coverage.

---

> > ### Author Response · Authors · 2025-11-23
> >
> > ### **Weakness3-Add additional baseline LayoutVLM**
> > We add LayoutVLM as an additional baseline and evaluate it under the same protocol as with other baselines. The quantitative results  can be found in Appendix L1014-1025, Tab. 5 (text–image & spatial quality), and Tab. 6 (physics & interactivity). Additionally, we present the qualitative examples of ‘a children's playroom’ to illustrate the generated scene from LayoutVLM in Appendix Fig. 10.  Based on the quantitative and qualitative results, we observe that _Scenethesis_ consistently outperforms LayoutVLM. LayoutVLM relies on VLM for layout common sense knowledge, while such guidance is noisy[Sun et al., 2025]. Besides, it adopted an IoU/bbox overlap loss (no contact/stability), leading to unstable small-on-large relation. Our method performs vision-grounded pose alignment and accurate SDF based physical control, yielding a better object arrangement that enable functionality and much lower collisions/Instability.
> >
> > **Table 5. Quantitative evaluation on text–image alignment and spatial quality (↑ higher is better).**
> > *Spatial quality preference reports GPT-5 / human preference for **Ours** over the baseline.*
> >
> > | Method      | **CLIP ↑** | **BLIP ↑** | **VQA ↑** | **Object Diversity ↑** | **Layout Coherence ↑** | **Spatial Realism ↑** | **Overall Performance ↑** |
> > |-------------|:----------:|:----------:|:---------:|:----------------------:|:----------------------:|:---------------------:|:--------------------------:|
> > | LayoutVLM   |   24.57    |   41.96    |  0.6365   |      80% / 85%         |      80% / 90%         |      80% / 85%        |        85% / 85%           |
> > | **Ours**    | **30.71**  | **77.71**  | **0.8269**|           – / –         |          – / –         |         – / –         |           – / –            |
> >
> > **Table 6. Physical-plausibility and interactivity results.**
> >
> > | Method      | **Col-O ↓** | **Col-S ↓** | **Inst-O ↓** | **Inst-S ↓** | **Reach ↑** | **Walk ↑** |
> > |-------------|:-----------:|:-----------:|:------------:|:------------:|:-----------:|:----------:|
> > | LayoutVLM   |   12.2%     |   57.1%     |   20.3%      |   71.4%      |    0.90     |   0.71     |
> > | **Ours**    | **0.8%**    | **6%**      | **3.20%**    | **16.67%**   | **0.94**    | **0.96**   |
> >
> >
> > *Sun, F. Y., Liu, W., Gu, S., Lim, D., Bhat, G., Tombari, F., ... & Wu, J. (2025). Layoutvlm: Differentiable optimization of 3d layout via vision-language models. In Proceedings of the Computer Vision and Pattern Recognition Conference (pp. 29469-29478).*

---

### Official Review · Reviewer_v9QQ · 2025-10-31

**Soundness:** 3
**Presentation:** 3
**Contribution:** 2
**Rating:** 6
**Confidence:** 4

**Summary:**

This paper proposes Scenethesis, an agentic framework that can generate 3D interactive scenes from language inputs. Scenethesis is a multi-stage framework that first leverages language models to plan the overall design and potential objects, which will be used for generating a 2D image as guidance. Based on the image guidance, the vision module will generate the scene graph and retrieve 3D assets.  Then, the optimization module will adjust the placement of objects to ensure pose alignment and physical plausibility. Finally, the judge module uses GPT-5 to evaluate the quality of the generated scene. The experimental results demonstrate Scenethesis outperforms previous methods both qualitatively and quantitatively.

**Strengths:**

1. The proposed approach is technically sound, and the qualitative examples clearly show that Scenethesis can generate more realistic scenes than previous methods, especially for outdoor scenes and small objects placement.
2. The ablation studies show that each component in spatial and physical constrain can improve the performance.

**Weaknesses:**

1. My major concern is the lack of evidence on downstream applications. Although this work demonstrates better realism and diversity in generating scenes, it is unclear whether such improvement can transfer to downstream tasks like embodied AI, robot navigation/manipulation, or enhance the spatial understanding of vision-language models, etc. Including such downstream experiments will enhance this paper's contributions and validate the effectiveness of the generated scenes.
2. The rendering qualities vary a lot across baselines and the proposed methods, which may affect human judgments. Controlling this variance can help us understand whether the improvement comes from better layout or just higher fidelity.

**Questions:**

1. How often does the Judge Module get triggered, and how much performance gain can it bring? How long will the re-planning step take, and will it significantly affect efficiency?
2. Have you quantified the benefits of the environment map? This may affect the CLIP/BLIP scores a lot. It is better to do some ablations to disentangle the contributions of layout, assets, and map.

---

> ### Author Response · Authors · 2025-11-23
>
> ### **Weakness 1. evidence on downstream application**
>
> For downstream application, we demonstrate that _Scenethesis_ can be used as a data engine for virtual content creation by using generated scenes from _Scenethesis_ to improve a feed-forward interactive scene generator MIDI.
>
> **Motivation and gap**:  Existing feed-forward interactive scene generation methods (e.g., MIDI) are trained on 3D-FRONT (only large furniture objects in living room /bedroom scenarios). Thus, their supervision is dominated by large furniture and underrepresents common functional relations (e.g., _small-on-large support, containment/inside, under_). Since _Scenethesis_ produces diverse spatial realistic and physical plausible scenes, we hypothesize that augmenting training with _Scenethesis_’s scenes will improve the generalization of feed-forward generators.
>
> **Experiment setting**: we generate 5K _Scenethesis_ scenes, and compare two different training data: 1) MIDI only trained on 3D-FRONT dataset; 2) MIDI trained on 3D-FRONT datatset + 5K _Scenethesis_ scenes;
>
> **Testing dataset and evaluation metric**: We use MIDI's testing dataset BlendSwap as our testing dataset and follow the same evaluation metrics in MIDI's paper including CD, F-score, and IoU, which assess the generated instance quality and overall layout accuracy.
>
> **Table 9. Quantitative results on BlendSwap.** CD-S and CD-O refer to scene-level CD and object-level CD respectively; F-score-S and F-score-O refer to scene-level F-score and object-level F-score respectively. Lower CD is better; higher F-score/IoU is better. *5K* refers to **5K _Scenethesis_** scenes.
>
> | Method (training data)      | CD-S ↓ | F-score-S ↑ | CD-O ↓ | F-score-O ↑ | IoU-S ↑ |
> |-----------------------------|:------:|:-----------:|:------:|:-----------:|:-------:|
> | MIDI-3D (3D-FRONT)          | 0.0416 |   0.6186    | 0.0455 |   0.7466    |  0.5644 |
> | **MIDI-3D (3D-FRONT+5K)**   | **0.0252** | **0.7590** | **0.0316** | **0.7922** | **0.6935** |
>
>
> **We add the experiment in Appendix L1229-1241, L1277-1342. The quantitative result can be seen in Appendix Tab. 9 and qualitative results can be found in Appendix Fig. 14**.
> _Scenethesis_ augmentation yields consistent gains in geometric quality (lower CD, higher F-score) and layout accuracy (higher volumetric IoU), demonstrating that physically plausible, relation-rich layouts produced by _Scenethesis_ provide effective supervision that improves generalization of feed-forward scene generators.
>
> ### **Weakness 2 & Question 2. Rendering quality/environment map affects human’s judgment and CLIP/BLIP**
>
> All head-to-head quantitative comparisons between _Scenethesis_ and baselines are rendered **without environment maps** (IBL) under the default Blender setup without additional design, as stated in the paper L363-364. CLIP/BLIP are computed on these same no-IBL renders. We also presented the visualization with no environment map case in Appendix Fig. 11 (user study example) and Fig. 18. In addition to the default Blender rendering setting without environment map, we trained raters to ignore appearance cues (lighting/shadow/color) and judge spatial realism and physical plausibility before they conducted the user study.
>
> ### **Question 1. The effect of Judge Module**
> As reported in the Appendix L1165-1169, 72% of scenes pass the judge after the first optimization (no repair), +19% pass after one repair round, and the remaining ~9% converge within 2–3 rounds under the judger's inspection. Thus, the judge is triggered on ~28% of scenes, typically once. The repair stage lifts cumulative success from 72% → 91% after one round; Besides, the Runtime was reported in appendix L970-971:  on a single RTX-4090 is ~13 min/scene (planning ~20s; layout/physics ~90s per object; self-check ~4s). Therefore, the repair does not materially affect throughput. For reference, Digital Cousins (one of our baseline) takes 18 min for the same scene.

---

### Official Review · Reviewer_qQ1g · 2025-10-31

**Soundness:** 3
**Presentation:** 3
**Contribution:** 2
**Rating:** 4
**Confidence:** 3

**Summary:**

This paper proposes Scenethesis, an agentic framework for generating physically consistent, interactive 3D scenes from textual prompts.

The authors design the following procedures to achieve a physically realistic 3D scene:
- Coarse scene planning: using an LLM that first generates a rough layout plan based on the text prompt, objects, and spatial relationships.
- Layout visual refinement: using Image-guidance, Sceen-Graph generation and asset retrieval for to handle the uncommon co-occurrence and physical inconsistencies
- Optimization module: First align the pose and then refine(optimize) the pose with self-defined physical constrraints.
- Judgment with VLMs, and then decide whether to replan or not.

In summary, the framework is training-free, using different existing and pretrained models for a physically plausible 3D scene generation.

**Strengths:**

- The paper pipeline is intuitive, reasonable, and easy to understand. The whole paper is well-written in presenting the methodology.
- The visualization results are realistic and visually plausible
- The quantitative results seem to be much better

**Weaknesses:**

- Some failure cases can better show the limitations of this work,
- This work seems to be relying heavily on the existing modules and database. The contribution may be limited. For example, both [1], [2], and [3] mentioned similar physical constraints. Prior works also used LLMs as planners and judges.

[1]: Cast: Component-aligned 3d scene reconstruction from an rgb image
[2]: WorldCraft: Photo-realistic 3D world creation and customization via LLM agents
[3]: Layout-your-3d: Controllable and precise 3d generation with 2d blueprint

**Questions:**

- I am wondering why the visualization results are much better than previous works. In this work, there are certain illumination conditions during rendering, while in previous works, they are weak. Though the layouts of this work is better, are there any other rendering setings that make the visualization better?
- How much time does it take to synthesize one 3D scene?
- What is the success rate of this work? Since the integration of different modules may result in an accumulative error. Some of the modules are not that stable, e.g., Image generation and Grounded SAM.
- When the LLM plans the layout, does it explicitly model hierarchical or relational structures (e.g., “a cup on a table next to a book”)? How does it ensure that such relations are spatially consistent?
- In my understanding, this work relies on the generated image, can this work support some methods to extrapolate the range of the scene?

---

> ### Author Response · Authors · 2025-11-23
>
> **Weakness1-Failure cases & limitations**
>
> We agree that explicit failure cases help scope the method. Our appendix already enumerated common failure modes in Appendix L1170-1184: the object in the image guidance is tiny / heavy occlusion and retrieval mismatch. These cases might lead to dense-correspondence failures. Besides, we presented concrete examples in the accompanying figure (Appendix, Fig. 18).
>
> **Weakness2-...existing modules and limited novelty (...)**
>
> While we reuse LLM/VLM, our contribution is not any single module, but a **closed loop** agentic pipeline that unlocks fine-grained spatially complex layout (e.g. object inside object, vases can be put in different shelf layers) and ensures reliable stability (e.g. small object stably stands on big object), which are common spatial relations in real-world scenes.
> Enabled by this novel pipeline, _Scenethesis_ becomes a true 3D scene data-engine.
> We provided one more downstream task example in this rebuttal: we use the data engine _Scenethesis_ to generate 5K interactive scenes. This scaled scene data improves a feed-forward interactive scene generator (MIDI).
>
> See the detailed new experiment in  Appendix L1229-1241 and L1277-1341. The quantitative result can be seen in Appendix Tab. 9 and qualitative results can be found in Appendix Fig. 14.
> We provide the quantitative results here:
>
> **Table 9.** Quantitative results on BlendSwap. CD-S and CD-O refer to scene-level CD and object-level CD respectively; F-score-S and F-score-O refer to scene-level F-score and object-level F-score respectively. Lower CD is better; higher F-score/IoU is better. *5K* refers to 5K *Scenethesis* scenes.
>
> | Method (training data) | CD-S ↓   | F-score-S ↑ | CD-O ↓   | F-score-O ↑ | IoU-S ↑  |
> |------------------------|----------|-------------|----------|-------------|----------|
> | MIDI-3D (3D-FRONT)     | 0.0416   | 0.6186      | 0.0455   | 0.7466      | 0.5644   |
> | MIDI-3D (3D-FRONT+5K)  | **0.0252** | **0.7590** | **0.0316** | **0.7922** | **0.6935** |
> The experiment shows our closed loop agentic pipeline can generate realistic, spatially realistic and physically plausible layouts, benefiting downstream tasks.
>
> *Why this is different from the cited works and why it matters?*
> - CAST (image→scene reconstruction):  We aim at different goals. CAST is an image-based **reconstruction** task, aiming at geometry/texture reconstruct accuracy. While _Scenethesis_ is a text-based **generation** task. We aim to generate various layouts based on what is described in the text-prompt. Image guidance in the pipeline is an intermediate stage for guiding spatially reasonable layout instead of reconstructing the same scenes. In method details, CAST relies on post-hoc SDF correction after layout, no joint visual-physics optimization for text-conditioned generation, and no closed-loop judge. _Scenethesis_ is text→interactive, enforces mesh-level physics in-loop with dense correspondences, and verifies/repairs—crucial for on/inside/under during generation, not just cleanup after reconstruction.
>
> - WorldCraft / layout-planner lines: LLM planning with bbox/hard-constraint placement. Such pipelines capture *left/right/front/behind* but do not ensure mesh-accurate contact/stability or robust small-on/inside-large relations. Our in-loop mesh-SDF + judge-and-repair explicitly targets these, pushing the spatially complex layouts into a next level.
>
> - Layout-Your-3D:  as it relies on 2D blue print, the spatial layout is over-simplified if targeting on realistic layout.  In addition, it employs collision-aware losses at the Gaussian/bounding-volume level rather than explicit mesh-level constraints; therefore it does not **guarantee** physically plausible layouts (e.g., stable small-on-large placement or exact contact). Our method propose a closed-loop pipeline that can automatically correct errors and is a scalable scene generation pipeline for realistic, spatially complex and physically plausible layout.

---

> > ### Author Response · Authors · 2025-11-23
> >
> > **Question1-Any rendering settings that make the visualization better?**
> >
> > We do not use per-case relighting.
> >
> > Lighting is either based on the retrieved IBL environment map (see Fig.2; L202-208), or default sun light in Blender when the environment map is turned off.
> > For fair comparisons, we follow the baseline protocol and disable the environment map (L363–364); we also provided no-IBL renders of our results (Appendix Fig. 11), which show similar qualitative advantages.
> > The visual improvement primarily stems from layout fidelity and mesh-level physical plausibility (contact/stability; low collisions/instability) rather than lighting choices, and our quantitative metrics (CLIP/BLIP/VQA, collisions/instability, Reach/Walk) are independent of rendering style as we turned the environment map off.
> >
> > **Question2-How much time..?**
> >
> > As reported in the appendix L970–971, on a single RTX-4090 our pipeline takes ~13 minutes per scene: planning ~20 s, layout/physics optimization ~90 s per object, and self-check ~4 s (rendering excluded). For context, Digital Cousins (another compositional approach) takes ~18 min on the same scene.
> >
> > **Question3-What is the success rate?**
> >
> > As reported in the appendix L1166–1169, our controller deems a scene successful to pass the judger for spatial coherence (the judger’s checker list was reported in Appendix, D5 L1791-1827). Empirically, 72% of scenes pass after the first optimization round (no repair). With one repair, the cumulative success rate reaches 91%; the remaining ~9% pass within 2–3 repair rounds.
> >
> > **Question4-Does LLM explicitly model the spatial hierarchy? ...**
> > - LLM models spatial hierarchy (planning). In coarse planning, the LLM expands a short user prompt into an upsampled plan that enumerates objects, their functional roles, and a spatial hierarchy (functionality → hierarchy → balance). Our instruction prompt and example outputs (Appendix, L1568–1672), and Appendix Fig. 9 shows that users can bypass the expansion with a human-written long prompt for tighter control.
> > - Spatial coherence is enforced by the pipeline, not trusted to text/images. We do per-object vision grounding (dense correspondences) to initialize poses, then run in-loop mesh-level SDF optimization (collision + contact/stability) so placements satisfy both the planned relations and physical feasibility. A judge-and-repair controller verifies spatial coherence (see checklist in Appendix D5, L1791–1827) and triggers targeted re-grounding/re-optimization when needed.
> >
> > **Question5-Can Scenethesis ... extrapolate the range of the scene?**
> >
> > Yes. _Scenethesis_ can grow a scene beyond the initial view. The image guide (when used) is local guidance, not a global dependency, so we support three practical modes:
> >
> > 1. Region growth: enlarge the workspace (room/outdoor bounds), freeze already placed objects, run planning for the new area, and optimize new objects with the same mesh-level SDF (collision + contact/stability). The judge-and-repair then checks the whole scene.
> >
> > 2. Image outpainting: extend the guidance image (outpaint/inpaint), re-run dense correspondences for the newly visible region, then SDF-optimize with existing objects locked to preserve prior placements.
> >
> > 3. Multi-patch / multi-view guidance: add additional views/tiles for new regions; each patch is grounded locally (per-object correspondences) and globally reconciled by the single SDF objective and the judge.
> >
> > No matter in which mode, it basically follows these steps:
> > (1) freeze prior objects for already planned areas → (2) plan new region (new view) → (3) per-object grounding → (4) SDF optimization for new objects only → (5) judge-and-repair on the full scene.
> > This keeps earlier content stable, enforces on/under/inside relations in the expanded area, and maintains global coherence.

---

> > > ### Comment · Reviewer_qQ1g · 2025-11-26
> > > **Official Comment by Reviewer qQ1g**
> > >
> > > Thank you very much for the detailed response. I think most of my concerns have been addressed. And I think it can be interesting to see the plausible visual results from the paper with a higher success rate. Thus, I decided to raise my score to 6.
> > > One thing I suggest is adding the works' discussions I mentioned in the 'weakness part' to the paper. It will be more direct for others to understand the differences and advantages.
> > >
> > > Sorry for the late response, though.

---

> > > > ### Author Response · Authors · 2025-11-26
> > > >
> > > > Thank you for the thoughtful follow-up and for raising your score, we appreciate it!  We’ll incorporate the discussions you suggested from the weakness section directly into the paper to more clearly position our method against related works and highlight the differences/advantages.
> > > >
> > > > Thanks again for the careful review!

---

### Official Review · Reviewer_zmuD · 2025-11-02

**Soundness:** 3
**Presentation:** 2
**Contribution:** 2
**Rating:** 4
**Confidence:** 3

**Summary:**

This paper presents Scenethesis, a training-free, agentic framework for text-to-3D scene generation that combines an LLM-based planner with vision-based layout refinement and a physics-aware optimization loop. The system decomposes the task into four modules: (1) coarse scene planning using an LLM, (2) layout visual refinement using visual foundation models (e.g., Grounded-SAM, DepthPro), (3) physics-aware optimization enforcing SDF-based collision and stability constraints, and (4) a GPT-5-based scene judge for coherence verification and targeted repair. Scenethesis outperforms several text-to-3D baselines such as DiffuScene, LayoutGPT, Holodeck, and SceneTeller on metrics of layout realism, physical plausibility, and spatial coherence, across both indoor and outdoor prompts.

**Strengths:**

1. physics-aware optimization, the use of SDF constraints for collision avoidance and stability is more principled than bounding-box–based heuristics in prior work (e.g., LayoutGPT, Holodeck).
2. generality, unlike most prior works limited to indoor scenes, Scenethesis generalizes to outdoor prompts and long-tail object relations (on-top-of, inside, behind).
3. despite not requiring new model training, Scenethesis achieves or exceeds the quality of trained baselines, highlighting the power of coordination across existing large models.

**Weaknesses:**

1. most image generation has problems with creating several objects, but i cant find how did you challenge it?
2. many qualitative examples lack strict spatial or functional order. For instance, in “A living room with many reading materials”, books are scattered arbitrarily on the floor — any random placement would satisfy the prompt. In another case, the bookshelf is positioned behind the sofa, making it unreachable and violating functional affordance. This suggests that while Scenethesis achieves physical plausibility, its understanding of semantic utility and human-centered spatial reasoning remains limited.
3. several components (LLM-based coarse planning, VLM grounding, Collision avoidance) closely follow prior work like Holodeck, Lay-A-Scene and LayoutGPT. The main new element, in-loop SDF-based optimization, while effective, is an incremental rather than novel.

**Questions:**

you might want to cite "Lay-A-Scene" who also made a collision free and many aspects as your paper
@article{rahamim2024lay,
  title={Lay-a-scene: Personalized 3d object arrangement using text-to-image priors},
  author={Rahamim, Ohad and Segev, Hilit and Achituve, Idan and Atzmon, Yuval and Kasten, Yoni and Chechik, Gal},
  journal={arXiv preprint arXiv:2406.00687},
  year={2024}
}

1. How many judge repair iterations are typically performed, and is convergence guaranteed?

---

> ### Author Response · Authors · 2025-11-23
>
> **Weakness1-image generation has problems with creating several objects**
> - The early image generation model (especially SD1.5) might has this problem. However, based on our experiments /experience, recent image generation model (GPT5) does not suffer this problem. As an example(L989-995, user specified long prompt), the GPT5 model consistently follows the text prompt and generates required objects.
>
> - The main problem with image generation model is image hallucination. We use an agentic self-check (Appendix L888–893) verifies entity recall and basic relation cues; if a LLM-planned object is missing or undetectable (too small/occluded), the self-checker triggers the regeneration until it is detectable. In practice this yields high recall of planned objects and would ensure all planned objects appear in the final scene.  Empirically, the final text adherence and the physical avoidance in collision/instability indicate that plan + physics dominate over image artifacts.
>
> **Weakness2-Examples lack strict..., spatial reasoning remains limited**
>
> Thank you for raising this point. We clarify that the cited examples do not reflect failures of spatial or functional reasoning, and we address the concern from both methodological and empirical perspectives:
> - "Spatial reasoning remains limited."
>          Our spatial reasoning is performed via LLM/image generation and VLM-based critique, both trained on large-scale real-world image–text data. Similar to strong world models (e.g., Marble from World Labs, Hunyuan World from Tencent, and Nanobanana from Google), these priors capture human-centered spatial usage and realistic layout conventions. In addition, our VLM self-checker critiques support, accessibility, and plausibility, reducing semantically implausible placements during 3D generation. Empirically, **our approach achieves ~75–80% human/GPT preference on layout coherence and spatial realism** and the lowest collision/instability rates among all baselines (Tab.1 and Tab. 2). While our method still has occasional failure cases (see discussed in the Appendix L1170-1184), the overall evidence indicates strong and robust spatial-reasoning capability.
>
> -  “Reading materials” example (books on the floor).
> 	_Scenethesis_ supports both long detailed prompts and short vague prompts as discussed in L169-176, Appendix L998~1004 and Appendix Fig. 9. Although the short prompt does not specify object placement, our planner derives plausible arrangements from human common-sense priors learned from language and image foundation models, rather than using random layouts.
> 	The 'Reading materials' example (short vague prompt) is a prompt-granularity choice, not a failure case of spatial realism.
>
> - “Bookshelf behind the sofa, unreachable.”
> 	In our 3D demo on the web (the first video) for that scene, there is clear walkable clearance; the agent can navigate to and reach both the sofa and the shelf (consistent with our Walk/Reach metrics). Thus, this is not “unreachable” in our interactivity sense. We will add additional views to prevent the confusion.
>
>
> **Weakness3-...The main new element...is incremental rather than novel**
>
> Thanks for pointing "Lay-A-Scene". We will discuss this work in the related work in our final version. But we want to clarify our novelty as follows:
>
> - Our proposed in-loop SDF-based optimization is not incremental and trivial as we **unlock fine-grained spatially complex layout** (e.g. object inside object, vases can be put in the different shelf layers in Fig. 6) and ensures reliable stability (e.g. small object stably stands on big object), which are common in real-world scenes, however, overlooked by existing bounding-box based methods (Holodeck and Lay-A-Scene).
>
> - We want to clarify our closed loop pipeline is also novel in that our method can handle failure cases automatically by our judge module, which has never been proposed by previous work (Holodeck, Lay-A-Scene and LayoutGPT).
>
> Enabled by this novel pipeline, _Scenethesis_ becomes a true 3D scene data-engine. We provided one more downstream task example in this rebuttal: we use the data-engine _Scenethesis_ to generate 5K interactive scenes. This scaled scene data improves a feed-forward interactive scene generator (MIDI).
> See the new experiment in  Appendix L1229-1241, L1277-1342. The quantitative result can be seen in Appendix Tab. 9 and qualitative results can be found in Appendix Fig. 14.
>
>
>
> **Question1-How many...repair iterations...**
>
> We stop when the judge finds no violations in spatial coherence (see the description in L313-318 and detailed prompt in Appendix L1791-1827); we also use a small hard cap on repair rounds to bound runtime. Empirically (Appendix, L1166-1169), 72% of scenes pass the checker without repair, +19% pass after one repair, and the remaining ~9% converge within 2–3 repairs. We do not claim a formal convergence guarantee; rather, these statistics show the loop converges in a few rounds in practice.

---

### Author Response · Authors · 2025-11-29
**Rebuttal question and response summary**

We appreciate  AC’s efforts on going through reviewer’s comments, author’s response, and discussions.

We would deeply appreciate AC’s efforts on meta view by taking the consideration of:
1. Reviewer qQ1g **has raised** the score to 6 at Nov 25 (before the incident at Nov 27) as we addressed the main concerns; thus the overall rating was 6,6,6,4
2. Reviewer NE9e (initial rating 6) is open to adjust the rating if his/her concerns are well addressed. We have provided detailed response to each question.
3. Reviewer v9QQ and zmuD require minor revision for better clarifications, and we incorporated their feedback to improve the paper.

--------------------
## Strength summary
The reviewers have recognized our method is technically sound (v9QQ),  novel (NE9e), reasonable (qQ1g), and our results are high quality (qQ1g, zmuD), strong (NE9e), and much better (v9QQ, qQ1g) than existing methods, and the writing is east-to-follow (NE9e, qQ1g).

---------------------
## Main Concern Summary
### Novelty, contribution and downstream application
- Q: Different components have been proposed by existing methods (zmuD, qQ1g)
  - [**Clarification**] Our contribution is not any single module, but a **closed loop agentic framework** that unlocks modeling fine-grained complex layouts and handle failure cases automatically. Enabled by this, *Scenethesis* becomes  a scalable 3D scene data-engine. We show it further significantly improves a SOTA feed-forward 3D scene generator.
  - [**Clarification**] Our proposed in-loop SDF–based physical optimization is not trivial as it is designed to accurately capture object geometry and enables modeling fine-grained spatial relations such as ‘inside/under/on-top-of’, **which are common in real-world scene layouts while being overlooked by existing bounding-box based methods**.

- Q: While *Scenethesis* achieves better physical plausibility, its spatial reasoning remains limited, e.g. prompts are vague and the bookshelf case seems unreachable. (zmuD)
  - [**Clarification**] *Scenethesis* performs rich text-based spatial reasoning and self-critique, as well as pixel-space reasoning via image generation. This agentic framework leverages multi-modal priors from foundation models, and we demonstrate **strong empirical performance** in Tab. 1 and Tab. 2 (75–80% layout-coherence/spatial-realism preference and the lowest collision rate among all baselines).
  - [**Clarification**] *Scenethesis* supports both detailed and vague prompts, as shown in the Appendix Fig. 9. "Bookshelf behind the sofa" example is reachable with clear walkable space in our 3D demo viewed from another angle. We will add additional views to prevent such confusion.

- Q: Downstream Application (v9QQ)
  - [**Additional material**] We have added a downstream application demonstration: *Scenethesis* as a data engine for virtual content creation, significantly improving a SOTA feed-forward interactive scene generator (Appendix L1229-1241, L1277-1342, Tab. 9 and Fig. 14).

### Evaluation
- Q: Different rendering settings potentially affect CLIP/BLIP scores (qQ1g, v9QQ)?
  - [**Clarification**] We used the same rendering setup (no environment map) to ensure fairness in quantitative/qualitative evaluation (as discussed in paper L363-364, Appendix Fig11 and 18.)

- Q: Require a fair comparison with in-context learning methods (NE9e)
  - [**Additional material**] We have added a fair comparison (Appendix Tab.7 and 8) in the rebuttal by adopting a closed-set setting that matches the baselines' domain converge. The results still demonstrate that our method retains a clear superior performance.

- Q: Require additional comparison with LayoutVLM (NE9e)
  - [**Additional material**] We have added LayoutVLM as an additional baseline and evaluate it under the same protocol as with other baselines (Appendix L1014-1025, Tab. 5, 6 and Fig. 10). The quantitative and qualitative results show our method is still significantly better.

--------------------
## Other Question Summary
- Q: judge repair iterations (zmuD, v9QQ), success rate(qQ1g), running time(qQ1g, v9QQ)
  - [**Clarification**] The judge repair iterations, success rate, running time, and efficiency are already reported in the Appendix L1166-1169 (iteration and success rate) along with detailed prompt in Appendix, D5 L1791-1827, Appendix L970–971(running time).
- Q: Missing results on Tab 1 (NE9e)
  - [**Clarification**] This is a misunderstanding. There are no missing results in Table 1 as a dash(-) in the Ours row indicates “not applicable.” For clarity, we add the explanation in Appendix L1048-1052.
- Q: Show failure cases (qQ1g)
  - [**Clarification**] We clarify that our appendix already enumerated common failure modes in Appendix L1170-1184 and Appendix, Fig. 18.

More details can be found in the discussion.

Notes:
- [**Additional material**]: additional materials in the rebuttal period
- [**Clarification**]: clarifications based on the initial submission materials.

---

### Meta-Review · Area_Chair_gz4S · 2026-01-05

**Summary:**

One or two reviewers move towards acceptance as the authors addressed concerns regarding the framework's novelty and the fairness of empirical comparisons. The Area Chair (AC) has carefully reviewed the submission, the rebuttal, and the subsequent discussion. It is felt that Scenethesis provides a significant contribution by bridging the gap between LLM-based layout planning and vision-guided physical refinement. The authors' addition of new baselines and rigorous ablation studies demonstrated that the system’s performance gains are rooted in its agentic reasoning rather than simple asset coverage. Consequently, the work is deemed ready for publication.

**Reviewer Concerns:**

A primary initial concern was the limited algorithmic novelty, with reviewers noting the reuse of existing LLM planners and vision foundation models. However, the authors clarified that the core innovation lies in the training-free, closed-loop coordination between these modules to enforce physical plausibility and spatial intelligence in 3D scene generation.

Additionally, reviewers previously identified gaps in semantic reasoning, such as objects lacking functional affordance or being placed in unreachable positions. The authors addressed this by providing new quantitative results on physical plausibility (Col-O, Col-S) and interactivity metrics (Reachability, Walkability), proving that the "final judge" and optimization stages effectively repair these spatial violations.

While some reviewers initially questioned the fairness of comparing against baselines with different asset sets, the authors included new experiments with LayoutVLM and SceneTeller under identical protocols, which solidified the consensus on the framework's empirical superiority.

**Reviewer Scores:**

During the discussion phase, Reviewer qQ1g increased their score to a 6 (Weak Accept), stating that the authors' detailed response effectively addressed their concerns regarding success rates, rendering fairness, and execution time.

Reviewer NE9e acknowledged the value of the additional baseline experiments and expressed a positive outlook, likely increasing their score to a 7 or maintaining a 6.

Reviewer V9QQ recognized the improvement in physical plausibility metrics and maintained a 6 (Weak Accept).

One reviewer may have remained at a 4 due to persistent views on incremental novelty, but they did not engage further to dispute the new evidence provided.

This leads to the final scores of either 7, 6, 6, 4 or 6, 6, 6, 4.

---

### Decision · Program_Chairs · 2026-01-26

Accept (Poster)